# Response of Rhizosphere Bacterial Communities to Near-Natural Forest Management and Tree Species within Chinese Fir Plantations

Jie Lei,[a] Hanbin Wu,[a] Xiaoyan Li,[a] Wenfu Guo,[b] Aiguo Duan,[a,c] Jianguo Zhang[a,c]

[a]State Key Laboratory of Tree Genetics and Breeding, Key Laboratory of Tree Breeding and Cultivation of the State Forestry Administration, Research Institute of Forestry, Chinese Academy of Forestry, Beijing, People's Republic of China
[b]Experimental Center of Tropical Forestry, Chinese Academy of Forestry, Pingxiang, People's Republic of China
[c]Collaborative Innovation Center of Sustainable Forestry in Southern China, Nanjing Forestry University, Nanjing, People's Republic of China

**ABSTRACT**   Near-natural forest management plays an important role in the maintenance of the long-term productivity and soil fertility of plantations. We conducted high-throughput absolute quantitative sequencing of 16S rRNA genes to compare the structures and diversity of rhizosphere soil bacterial communities among a pure Chinese fir (*Cunninghamia lanceolata*) plantation (S), a *Cunninghamia lanceolata-Castanopsis hystrix-Michelia hedyosperma* mixed plantation (SHX), and a *Cunninghamia lanceolata-Castanopsis fissa* mixed plantation (SD). The results revealed that near-natural forest management improved the rhizosphere soil properties of Chinese fir, especially the phosphorus content. Rhizosphere soil bacterial communities of Chinese fir in SHX and SD contained higher total absolute abundances and more unique operational taxonomic units (OTUs) than the pure plantation forest. *Planctomycetes* and *Actinobacteria* were abundant in SD, and *Actinobacteria* were enriched in SHX. The tree species also had an impact on the rhizosphere soil bacterial communities. For the rhizosphere soils of different tree species of SHX, the available phosphorus (AP) content of the rhizosphere of Chinese fir significantly surpassed those of *Castanopsis hystrix* and *Michelia hedyosperma*. Bacteria related to nitrogen fixing, such as *Burkholderiales* and *Rhizobiales*, were more abundant in Chinese fir in SD than in *Castanopsis fissa*. *Acidobacteria* and *Proteobacteria* underpinned the differences found in the compositions of soil bacteria. The pH and soil organic matter were key variables influencing the rhizosphere soil bacterial communities. Our results demonstrated that in Chinese fir plantations, 12 years of near-natural management of introduced broad-leaved tree species can drive alterations of the physicochemical characteristics, bacterial community structure, and composition of rhizosphere soil, with tree species identity further influencing the rhizosphere soil bacterial community.

**IMPORTANCE**   Near-natural forest management is an important way to change the soil fertility decline and productivity reduction of pure Chinese fir plantations. At present, many detailed studies have been carried out on the impact of near-natural forest management on Chinese fir plantations at home and abroad. However, there are still few studies on the response of rhizosphere bacterial communities to near-natural forest management. Our study determined absolute quantities of Chinese fir rhizosphere bacterial communities in different mixed patterns. The results underscore the importance of near-natural forest management for Chinese fir plantation rhizosphere bacterial communities and provide new information on soil factors that affect rhizosphere bacterial communities in South China.

**KEYWORDS**   high-throughput sequencing, near-natural forest management, mixed forest, rhizosphere bacteria, soil properties, tree species

Address correspondence to Aiguo Duan, duanag@caf.ac.cn.

The authors declare no conflict of interest.

As a gateway for plants to absorb water and nutrients and to interact with the soil matrix, the rhizosphere zone plays a pivotal role in plant life and soil ecosystems (1). In nature, the interactions between roots and soil are very complex, in which rhizosphere microorganisms actively participate (2). The properties of rhizosphere soil are determined largely by the interactions of soil, plants, and rhizospheric microorganisms. Some rhizosphere microorganisms can fix nitrogen; solubilize phosphorus, potassium, as well as other micronutrients (3, 4); and help the root system effectively absorb these nutrients. Conversely, changes in key rhizosphere soil nutrients such as C, N, and P can significantly impact soil microbial communities and the restoration of vegetation (5, 6). The myriad interactions among them render the rhizosphere's soil properties, soil enzyme activities, and microorganisms significantly different from those of bulk soils (7, 8). Recently, N. Ren et al. (9) found that $N_{180}$ (N input of 180 kg N $hm^{-2}$) treatment significantly increased the abundance of *Actinobacteria* in soil. The long-term use of phosphate fertilizer has also been found to reduce the total operational taxonomic units (OTUs) and diversity of rhizosphere bacteria (10). In contrast, applying organic fertilizers increases the relative abundances of *Burkholderiales*, *Myxococcales*, and *Nitrospirales* (among others) in tea rhizosphere soil, helping to change the rhizosphere microorganisms' structure and recruit beneficial bacteria to the rhizosphere (11). Furthermore, soil type, plant growth stage, farming practices, and other environmental factors may shape the composition of the rhizosphere microbial community (12–14). In addition to the interaction between rhizosphere nutrients and rhizosphere microorganisms, the structure and functional diversity of rhizosphere microorganisms are often affected by plant species, due mainly to differences in root exudates and rhizosphere deposition in differing root zones (15–17). Rhizodeposition, which includes the shedding of root cells and the exudation and leakage of substances such as sugar, organic acids, and amino acids, can be utilized by microorganisms as substrates to increase their biomass and activity (18, 19).

The concept of "close to nature" was introduced in China in the late 1990s by Q. Shao (20). In near-natural forest management, procedures like selection cutting and natural regeneration of trees were implemented to create an uneven-aged forest distinguished by multiple tree species and multiple strata (21). To do this, forest gaps were first created in a pure artificial forest via high-intensity selective cutting, and other tree species were then interplanted under the forest canopy to form a near-natural mixed forest (22, 23). The transformation of the pure forest into a mixed forest generally changes the undergrowth vegetation and spatial structure of their stands (24–26). Compared with stands of a pure forest, those of a mixed plantation formed through near-natural forest management have increased biomasses in the arboreal, shrub, and herb layers (27–29) and a greater ecosystem carbon fixation capacity (30). Interplanting with different broad-leaved tree species under a *Pinus massoniana* Lamb. forest canopy improved the contents of total nitrogen, organic matter, alkali-hydrolyzed nitrogen, available phosphorus, and available potassium in the soil (31). Similarly, converting pure Chinese fir forests into mixed forests enhanced their nutrient content and ameliorated the soil properties (32, 33). Under near-natural forest management, the activity of soil microorganisms will inevitably be affected by the corresponding effects from aboveground plant parts. Numerous studies have investigated the impact of near-natural forest management on soil microorganisms (34–36). A recent study found that the soil microbial carbon and nitrogen contents of Chinese fir (*Cunninghamia lanceolata* [Lamb.] Hook.) tree stands under near-natural management exceeded those of a second-generation Chinese fir plantation (33). Compared with a pure plantation forest, the soil microbial diversity in terms of the Shannon index, the richness index, and the evenness index is increased in plantations after they undergo a near-natural transformation (37, 38).

Chinese fir is an important fast-growing and high-yield tree species in southern China. According to the results of the Ninth National Forest Resources Inventory in China, the area and forest stock volume of Chinese fir plantations accounted for 27.23% and 32.57% of the country's main dominant plantation species, ranking first in both respects (39). However, due to certain traits of Chinese fir and its cultivation measures, some problems arose, such as deteriorating soil properties and decreasing

**TABLE 1** Rhizosphere soil properties of Chinese fir trees in different mixed forests[a]

| Mixed pattern | Mean pH ± SD | Mean SOM concn (g·kg⁻¹) ± SD | Mean SOC concn (g·kg⁻¹) ± SD | Mean TN concn (g·kg⁻¹) ± SD | Mean AN concn (mg·kg⁻¹) ± SD | Mean TP concn (g·kg⁻¹) ± SD | Mean AP concn (mg·kg⁻¹) |
|---|---|---|---|---|---|---|---|
| S_S | 4.5 ± 0.14 A | 35.24 ± 1.56 A | 20.44 ± 0.91 A | 1.57 ± 0.05 A | 181.02 ± 11.67 A | 0.33 ± 0.01 B | 10.94 ± 1.58 A |
| SHX_S | 4.36 ± 0.17 A | 45.67 ± 11.3 9A | 26.50 ± 6.61 A | 1.87 ± 0.56 A | 223.31 ± 70.78 A | 0.47 ± 0.08 A | 6.50 ± 1.56 B |
| SD_S | 4.66 ± 0.14 A | 40.53 ± 6.85 A | 23.51 ± 3.97 A | 1.76 ± 0.36 A | 178.00 ± 28.57 A | 0.32 ± 0.03 B | 10.85 ± 2.79 A |

[a]Values are the means ± standard deviations ($n = 3$). Different letters indicate significant differences ($P < 0.05$). SOM, soil organic matter; TN, total nitrogen; TP, total phosphorus; AN, available nitrogen; AP, available phosphorus; S_S, Chinese fir in a pure *Cunninghamia lanceolata* forest; SHX_S, Chinese fir in a *Cunninghamia lanceolata-Castanopsis hystrix-Michelia hedyosperma* mixed forest; SD_S, Chinese fir in a *Cunninghamia lanceolata-Castanopsis fissa* mixed forest.

productivity, all of which are not conducive to the sustainable management of Chinese fir plantations (40–42). But through stand thinning and interplanting, transforming a pure coniferous plantation forest into a near-natural forest consisting of mixed coniferous and broad-leaved trees of different sizes and ages is gradually emerging as one of the most promising ways to replace large areas of coniferous monocultures in China (43). Yet current research on the near-natural forest management of Chinese fir plantations is limited mainly to its impacts on understory vegetation diversity, stand spatial structure, and stand growth (44–46). Therefore, in the present study, we used 16S rRNA genes to investigate the diversity and structure of the rhizosphere bacterial communities of different tree species. Our findings provide insight into the effects of stratified mixed patterns formed by near-natural forest management and tree species on the belowground bacterial communities and their relationships to soil nutrients under different mixed forests, which can serve as a practical basis for maintaining the soil fertility of Chinese fir plantations and securing their long-term productivity.

## RESULTS

**Chinese fir in different mixed forests. (i) Physicochemical properties of rhizosphere soil.** The rhizosphere soil properties of Chinese fir displayed pronounced changes under near-natural forest management (Table 1). The total phosphorus (TP) content was significantly higher, reaching 0.47 g kg⁻¹, in rhizosphere bacterial communities of Chinese fir in a *Cunninghamia lanceolata-Castanopsis hystrix-Michelia hedyosperma* mixed forest (SHX_S) than in either Chinese fir in a pure *Cunninghamia lanceolata* forest (S_S) or Chinese fir in a *Cunninghamia lanceolata-Castanopsis fissa* mixed forest (SD_S), but the reverse was found for the available phosphorus (AP) content, being only 6.5 mg kg⁻¹ in SHX_S. In contrast, the other five indicators (pH, soil organic matter [SOM], soil organic carbon [SOC], total nitrogen [TN], and available nitrogen [AN]) showed no significant differences among the three stand types ($P > 0.05$). Compared with S_S, the rhizosphere soil pH of SD_S was slightly higher, at pH 4.7 ± 0.1, whereas that of SHX_S was lower, at just pH 4.4 ± 0.1. These results revealed that the introduction of *Michelia hedyosperma* and *Castanopsis hystrix* reduced the contents of AP in Chinese fir rhizosphere soil.

**(ii) Composition and diversity of bacterial communities.** To investigate the diversity and structure of rhizosphere bacterial communities of Chinese fir in different mixed forests, we performed high-throughput absolute-abundance sequencing of 16S rRNA genes. The rhizobacterial OTUs of Chinese fir under mixed treatment increased by 1.08 to 1.12 times compared with pure forest, among which SHX_S had the highest value (Fig. 1c). More unique OTUs occurred in the rhizosphere soil of SHX_S and SD_S. Next, the taxonomic distributions of these bacterial OTUs were assessed (Fig. 1a and b). Whether at the phylum or the genus level, no significant differences in the compositions of bacterial communities were detected among the rhizosphere soils of Chinese fir trees growing in different mixed forests. However, the rhizosphere soil bacterial communities of Chinese fir in SHX and SD contained higher total absolute abundances than those in the pure forest (S), reaching 564,063 and 618,759, respectively. Seven phyla with absolute abundances of >1% were identified: *Acidobacteria, Proteobacteria, Actinobacteria, Chloroflexi, Planctomycetes, Gemmatimonadetes, Bacteroidetes,* and No_Rank (there is no clear taxonomic information or taxonomic name at a taxonomic level) (Fig. 1a). *Acidobacteria* and *Proteobacteria* dominated the bacterial communities in the rhizosphere of Chinese fir, accounting for 71.6% to 76.5% of the total absolute

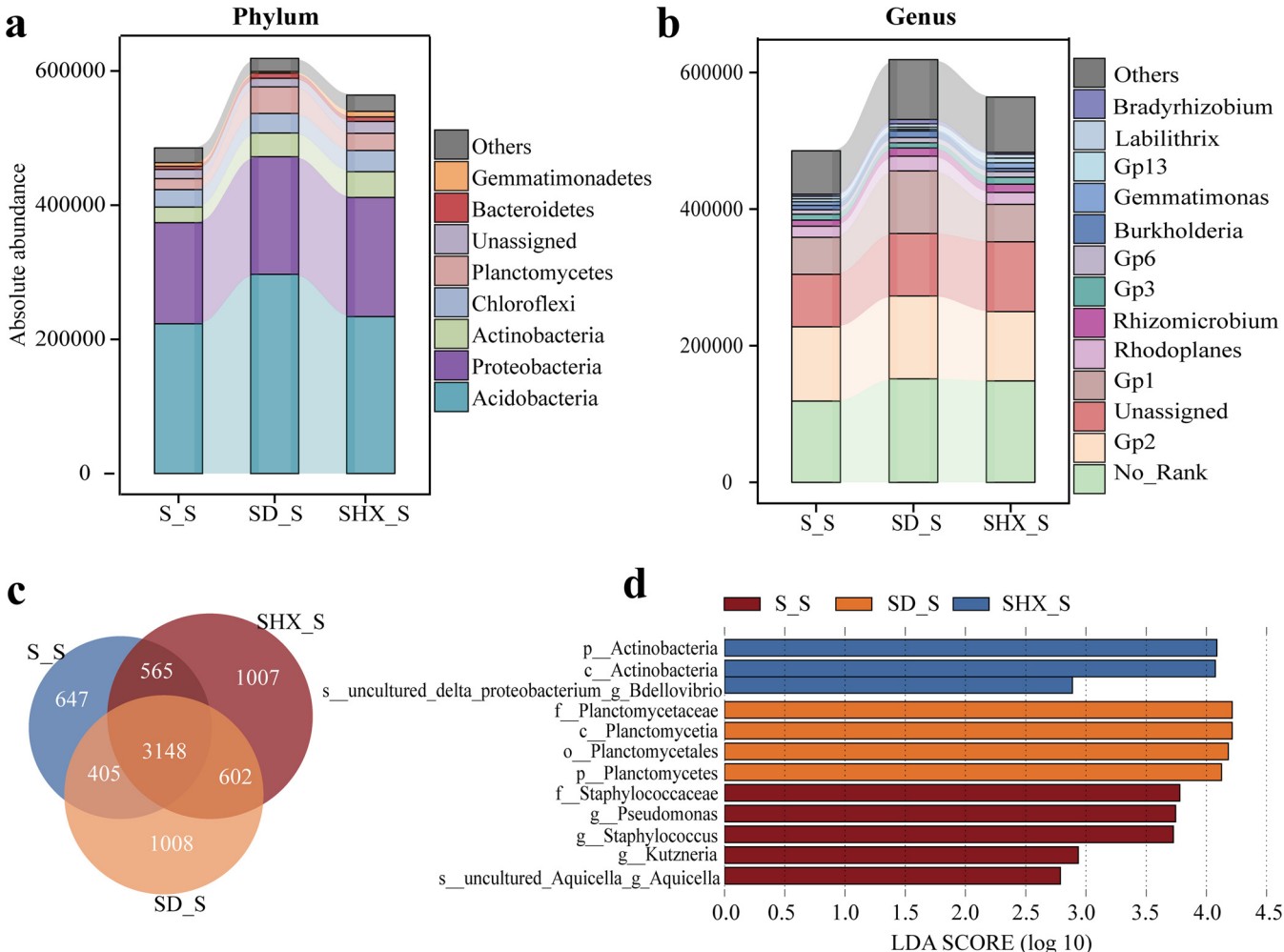

**FIG 1** Rhizosphere bacterial community composition and bacteria with significant differences in Chinese fir in different mixed patterns. (a) Rhizosphere bacterial community composition at the phylum level. (b) Rhizosphere bacterial community composition at the genus level. (c) Bacterial OTU distribution in the Chinese fir rhizosphere soil. (d) LDA scores of taxa enriched in Chinese fir in different mixed patterns. Only taxa with LDA values of >2.2 ($P < 0.05$) are shown.

abundance. At the genus level (Fig. 1b), 14 genera were detected at a proportion of >1%, of which No_Rank, Gp2 (*Acidobacteria* subgroup 2), unassigned, and Gp1 (*Acidobacteria* subgroup 1) were the prominent genera. The cumulative absolute abundances of the four genera accounted for 72.9% to 74.2% of the total absolute abundance. To determine the differences among species in the Chinese fir rhizosphere soil bacterial communities with different mixed patterns, we used both one-way analysis of variance (ANOVA) and the linear discriminant analysis (LDA) effect size (LEFSE) algorithm (LDA log score of >3.0 and $P$ value of <0.05) (Fig. 1d [the length of the bar chart represents the impact of different species]; see also Fig. S1 in the supplemental material). The one-way ANOVA results showed that the absolute abundances of *Planctomycetes*, *Actinobacteria*, *Clostridium*, *Novosphingobium*, and *Pseudonocardia* were significantly higher in SD_S than in S_S ($P < 0.05$) (Fig. S1). *Verrucomicrobia* and *Actinobacteria* were more abundant in SHX_S than in S_S, while the absolute abundances of *Staphylococcus* and *Rhodococcus* were significantly higher in S_S than in SHX_S and SD_S, respectively ($P < 0.05$). The LDA bar chart shows that the family *Staphylococcaceae* had the highest LDA score (3.779) of rhizosphere bacteria in S_S, indicating that it was considerably enriched in S_S (Fig. 1d). *Actinobacteria* (4.086) had the highest LDA scores and were considerably enriched in SHX_S, whereas *Planctomycetaceae* (4.803) were significantly enriched in SD_S.

Alpha diversity indices were analyzed in all samples to estimate the richness of the

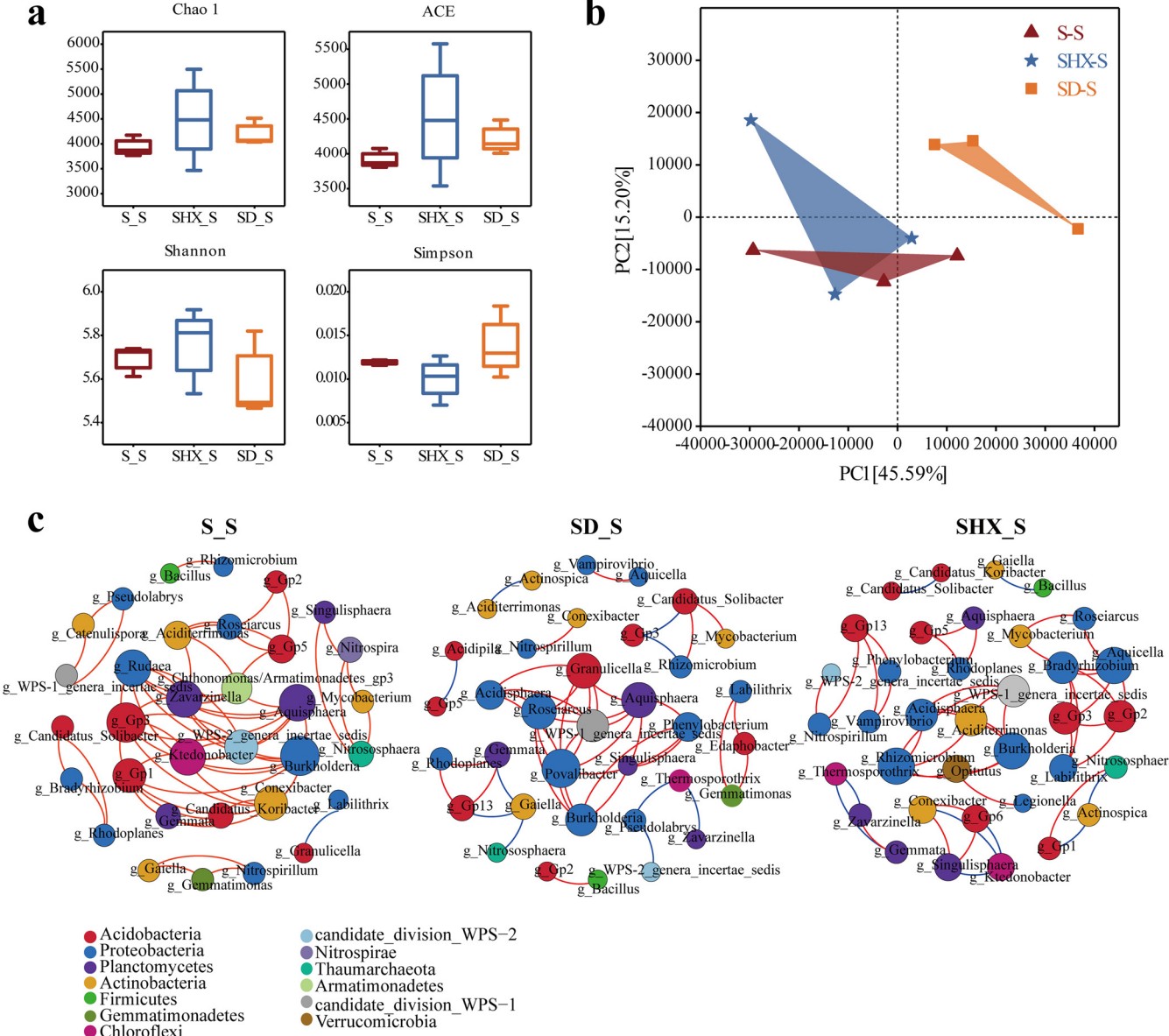

**FIG 2** Bacterial diversity and bacterial interaction networks in Chinese fir in different mixed patterns. (a) Alpha diversity of rhizosphere bacterial communities. (b) Principal-component analysis (PCA) of bacterial communities in the rhizosphere of Chinese fir. (c) Interaction network of dominant members of the microbiota at the genus level (top 40) in rhizosphere soil of S_S, SD_S, and SHX_S. The different colors indicate the corresponding taxonomic assignments at the phylum level. The edge colors represent positive (red) and negative (blue) correlations. Only significant interactions are shown ($r > 0.6$; $P < 0.05$).

rhizosphere bacterial communities (Fig. 2a). Except for the Simpson index, the three other diversity indices of the rhizosphere bacterial community were highest in SHX_S. Changes in the diversity indices from the pure forest to the mixed forest revealed that near-natural forest management had improved the species diversity and uniformity of the rhizosphere bacterial community of Chinese fir. Principal-component analysis (PCA) showed that 60.78% of the variance was explained by the first two components (Fig. 2b). The separation of the SD_S samples from those of S_S and SHX_S demonstrated a great difference between SD_S and the other two stand types.

Network analysis of the top 40 bacterial species at the genus level showed that the Chinese fir trees of the mixed forest harbored more nodes and fewer edges than those of the pure forest (Fig. 2c and Table S1). These results indicated that the interaction network in SD_S and SHX_S was relatively simple compared to that in the pure forest,

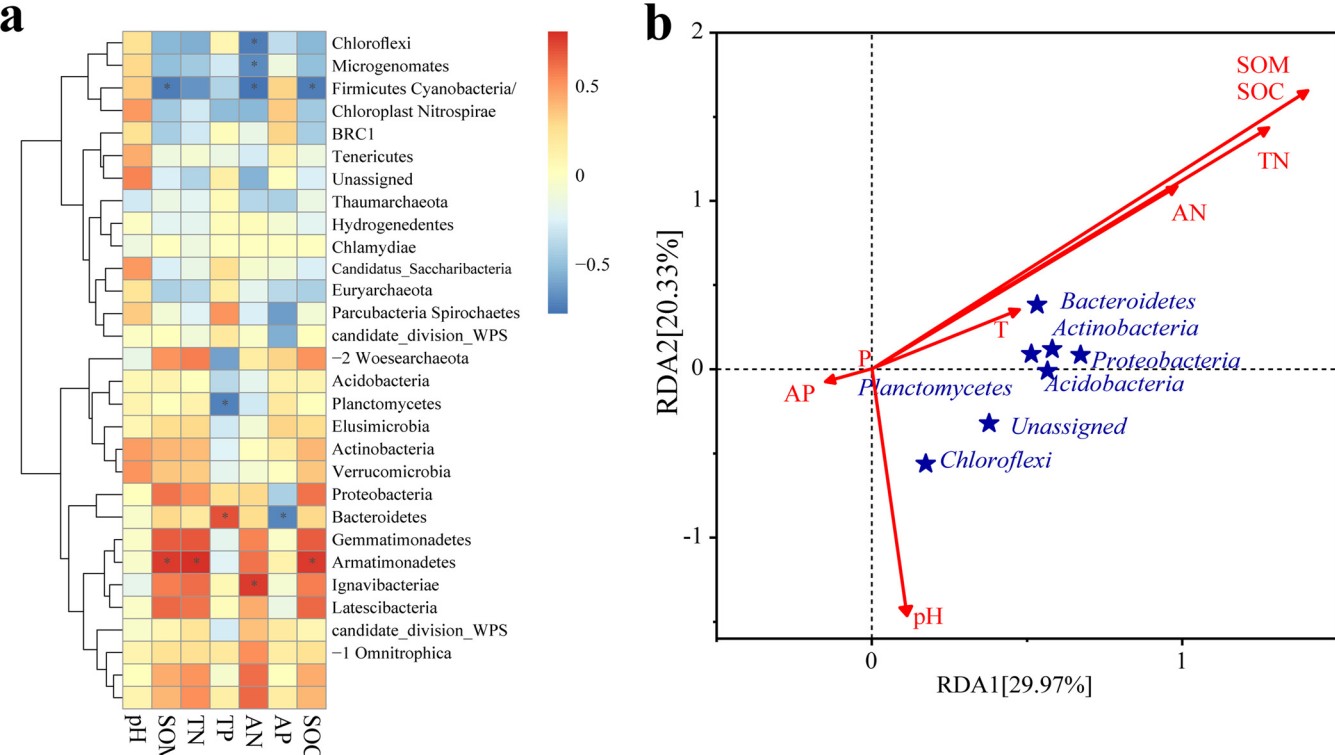

**FIG 3** Correlation relationships (a) and redundancy analysis (RDA) (b) of Chinese fir rhizosphere bacteria and rhizosphere soil physicochemical properties in different plantation stand types at the phylum level. *, $P < 0.05$.

but the complexity between bacteria was elevated. The key microbial groups observed in this study belonged mainly to *Acidobacteria*, *Proteobacteria*, *Planctomycetes*, and *Actinobacteria*, and the centrality of *Acidobacteria* became lower and that of *Proteobacteria* became higher in SD_S and SHX_S than in S_S.

**(iii) Relationships between soil physicochemical properties and bacterial community structure.** SOM, TN, and SOC had significant positive associations with *Bacteroidetes* ($P < 0.05$) according to a heat map of the tested correlations between the absolute abundances of rhizosphere bacterial communities and soil physicochemical parameters (Fig. 3a). AN was significantly positively correlated with *Gemmatimonadetes* yet negatively correlated with *Chloroflexi* ($P < 0.05$). Although TP had a significant positive correlation with *Verrucomicrobia*, AP had a significant negative correlation with it ($P < 0.05$). Redundancy analysis (RDA) at the phylum level (Fig. 3b) demonstrated that SOM, TN, AN, TP, and pH values were the main rhizosphere soil property factors that significantly impacted the soil rhizosphere microbes. RDA axes 1 and 2 captured 29.97% and 20.33% of the variability in rhizosphere soil bacteria, respectively. At the phylum level, SOM exerted the most substantial effect on the bacterial community in the rhizosphere of Chinese fir.

**Different tree species in the same mixed forest. (i) Physicochemical properties of rhizosphere soil.** As shown in Table 2, the AP content of rhizosphere soil in SHX_S was significantly higher than those of *Castanopsis hystrix* in SHX (SHX_H) and *Michelia hedyosperma* in SHX (SHX_X) (by 2.93 and 2.84 mg·kg$^{-1}$, respectively). The pH in the rhizosphere soil of SHX_X was the highest, reaching 4.60, and that of SHX_S was the lowest, at only 4.36. For the contents of SOM, SOC, TN, TP, and AN, each exhibited a trend of SHX_S > SHX_H > SHX_X, whereas the trend for pH shifted to SHX_S < SHX_H < SHX_X. Except for pH and TP, the SOM, SOC, TN, AN, and AP contents in the rhizosphere soil were all lower in SD_S than in SD_D. The pH value of SD_D was 4.47, which was lower than that of SD_S (pH 4.66).

**(ii) Composition and diversity of bacterial communities.** The bacterial communities of SHX_S, SHX_H, and SHX_X contained 5,322, 5,619, and 5,263 OTUs, respectively (Fig. 4c).

**TABLE 2** Rhizosphere soil physicochemical properties of different tree species in SHX and SD[a]

| Tree species | Mean pH ± SD | Mean SOM concn (g·kg$^{-1}$) ± SD | Mean SOC concn (g·kg$^{-1}$) ± SD | Mean TN concn (g·kg$^{-1}$) ± SD | Mean AN concn (mg·kg$^{-1}$) ± SD | Mean TP concn (g·kg$^{-1}$) ± SD | Mean AP concn (mg·kg$^{-1}$) |
|---|---|---|---|---|---|---|---|
| SHX_S | 4.36 ± 0.17 A | 45.67 ± 11.39 A | 26.50 ± 6.61 A | 1.87 ± 0.56 A | 223.31 ± 70.78 A | 0.47 ± 0.08 A | 6.50 ± 1.56 A |
| SHX_H | 4.40 ± 0.12 A | 36.27 ± 7.59 A | 21.04 ± 4.4 A | 1.58 ± 0.21 A | 192.67 ± 25.38 A | 0.46 ± 0.07 A | 3.56 ± 0.29 B |
| SHX_X | 4.60 ± 0.14 A | 33.81 ± 2.55 A | 19.61 ± 1.48 A | 1.26 ± 0.28 A | 186.63 ± 24.37 A | 0.43 ± 0.07 A | 3.66 ± 0.65 B |
| SD_S | 4.66 ± 0.14 A | 40.53 ± 6.85 A | 23.51 ± 3.97 A | 1.76 ± 0.36 A | 178.00 ± 28.57 A | 0.32 ± 0.03 A | 10.85 ± 2.79 A |
| SD_D | 4.47 ± 0.09 A | 42.93 ± 3.76 A | 24.90 ± 2.18 A | 1.90 ± 0.05 A | 203.03 ± 23.02 A | 0.32 ± 0.02 A | 13.69 ± 0.87 A |

[a]Values are the means ± standard deviations ($n = 3$). Different letters indicate significant differences ($P < 0.05$). SHX_S, Chinese fir in SHX; SHX_H, *Castanopsis hystrix* in SHX; SHX_X, *Michelia hedyosperma* in SHX; SD_S, Chinese fir in SD; SD_D, *Castanopsis fissa* in SD.

Overall, 3,603 OTUs were shared by the three tree species, whereas 800, 860, and 569 OTUs were unique to the rhizospheres of SHX_S, SHX_X, and SHX_H, respectively. The total absolute abundances of the bacterial communities of SHX_S were 181.74% and 148.31% higher than those of SHX_X and SHX_H, respectively (Fig. 4a). At the phylum level, seven phyla with a proportion of >1% were detected, namely, *Acidobacteria*, *Proteobacteria*, and *Chloroflexi*, etc. At the genus level, a total of 13 genera with a proportion of >1% in the sample were detected (Fig. 4b). Among them, the dominant ones were unassigned, Gp2, Gp1, and No_Rank, and the cumulative absolute abundances of these four genera constituted 72.0% to 72.9% of the total absolute abundance.

There were 5,163 and 5,308 bacterial OTUs in the rhizosphere soil samples of SD_D and SD_S, respectively (Fig. 5c). The bacterial communities of SD_S and SD_D had 1,315 and 1,170 unique OTUs, respectively. As evidenced by the data in Fig. 5a and b, the compositions of the rhizosphere bacterial communities of SD_S and SD_D were similar. *Acidobacteria* and *Proteobacteria* were the dominant phyla in SD_D and SD_S, with the former having the highest absolute abundance. The cumulative absolute abundances of *Acidobacteria* and *Proteobacteria* were 472,340 and 675,679, accounting for 76.34% and 72.95% of the total absolute abundance, respectively. The absolute abundance of SD_D was significantly higher than that of SD_S, but the absolute abundance of SD_S was only 0.67 times higher than that of SD_D (Fig. 5a). At the genus level, 10 genera with a proportion of >1% were detected. Of these, four genera belonged to *Acidobacteria*, namely, Gp2, Gp1, Gp6, and Gp3. The absolute abundances for the same genus were ranked as SD_S < SD_D.

The results of one-way ANOVA (Fig. S2) indicated that among the bacterial phyla and genera with significant differences in SHX ($P < 0.05$), the absolute abundances of *Actinobacteria*, Gp1, *Kitasatospora*, *Paenibacillus*, *Rhodoplanes*, and *Streptomyces* in SHX_S were significantly higher than those in SHX_H and SHX_X. In contrast, the absolute abundances of *Thaumarchaeota*, *Gemmata*, Gp7, *Longilinea*, and *Longispora* in the rhizosphere soil of SHX_X were the highest. As for SD, it can be seen that the phyla and genera with significant differences, such as *Proteobacteria*, *Planctomycetes*, *Dactylosporangium*, *Gaiella*, Gp1, *Granulicella*, Gp3, *Novosphingobium*, *Terriglobus*, *Rickettsia*, *Roseiarcus*, and *Streptacidiphilus*, all exhibited higher absolute abundances in SD_D than in SD_S (Fig. S3).

We calculated the LDA values among different tree species in SHX and SD. The results demonstrated that in SHX, *Anaerolineaceae* (*Chloroflexi*) were enriched in SHX_X, with the highest LDA score (3.571); *Actinobacteria* were enriched in SHX_S, with an LDA value of 4.124. The enriched genus in SHX_H was *Salmonella*, with an LDA score of up to 3.202 (Fig. 4d). In SD, *Proteobacteria* and *Actinobacteria* were predominantly enriched in SD_D, with more differentially significant taxa, including *Roseiarcus*, *Rhizobiales*, *Sphingomonadaceae*, *Oxalobacteraceae*, *Comamonadaceae*, *Hyphomicrobiaceae*, *Burkholderiales* (*Proteobacteria*), *Gaiella*, *Jatrophihabitans*, and *Streptacidiphilus* (*Actinobacteria*). *Telmatospirillum* (*Proteobacteria*) was enriched in SD_S (Fig. 5d).

The species richness of the rhizosphere bacteria in SHX_H exceeded that of SHX_S, whereas that of SHX_X was the lowest (Fig. 6a). The Shannon index of SHX_X was the highest, indicating that its rhizosphere bacterial community had the greatest diversity. PCA of the first two components explained 71.36% distribution of rhizosphere bacteria (Fig. 6b). Rhizosphere bacteria of SHX_H and SHX_X were more similar in their rhizosphere bacterial community compositions. For different tree species in SD, excluding the Shannon index, the Chao1, abundance-based coverage estimator (ACE), and Simpson indices for the rhizosphere

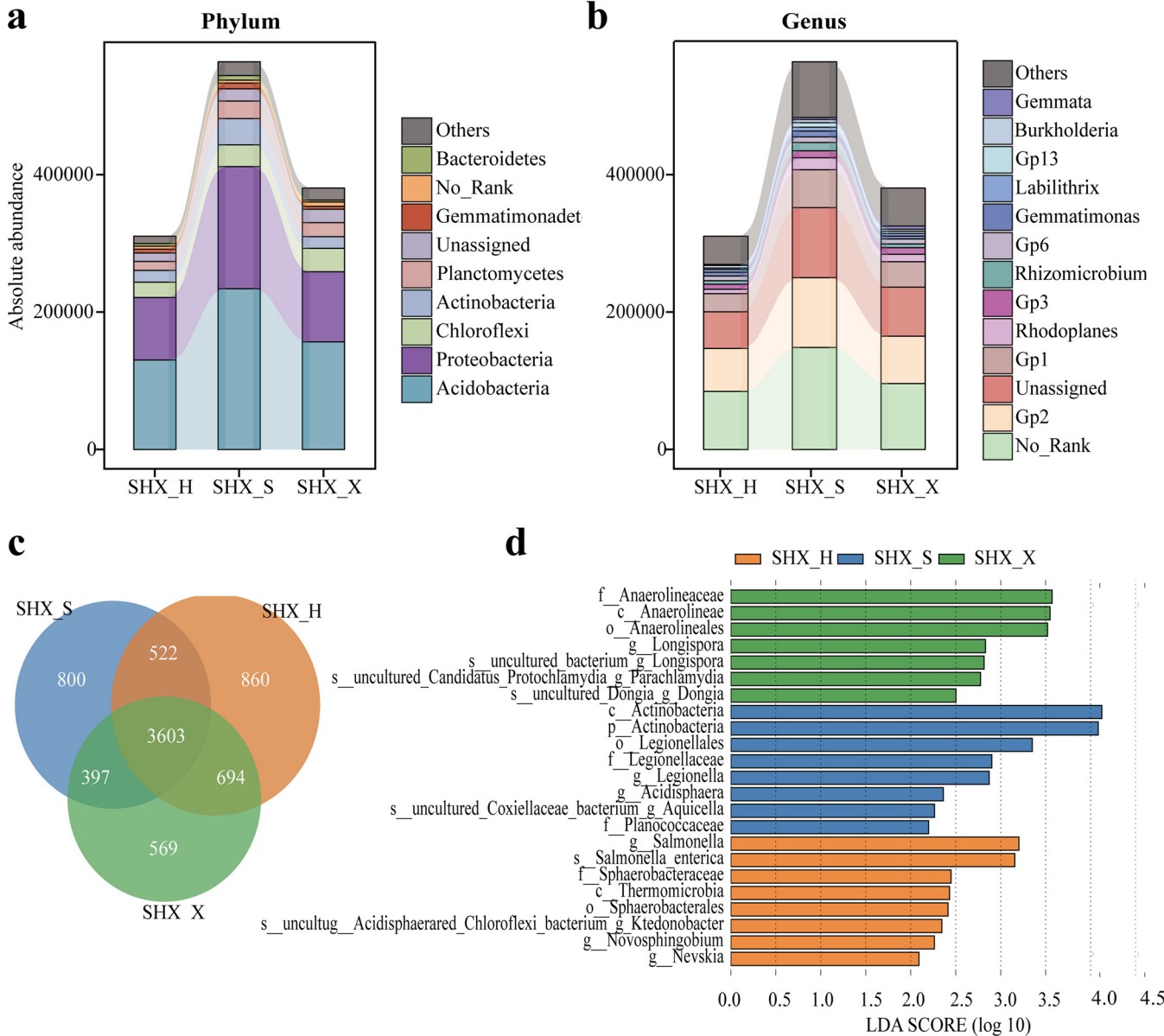

**FIG 4** Rhizosphere bacterial community composition and bacteria with significant differences in different tree species at SHX. (a) Rhizosphere bacterial community composition at the phylum level. (b) Rhizosphere bacterial community composition at the genus level. (c) Bacterial OTU distribution in rhizosphere soil. (d) LDA scores of taxa enriched in different tree species at SHX. Only taxa with LDA scores of >2.2 ($P < 0.05$) are shown.

bacteria of SD_S were all higher than those of SD_D, indicating that the former's diversity of rhizosphere bacteria was less than the latter's (Fig. 7a). PCA confirmed that the samples from these two groups showed a clear separation (Fig. 7b). In the network analysis of the absolute abundances of the top 40 bacteria at the genus level in SHX, there were 36 nodes and 47 edges in SHX_S, 33 nodes and 38 edges in SHX_H, and 32 nodes and 40 edges in SHX_X (Fig. 6c and Table S2). Compared to SHX_S, the complexity of the interaction network in SHX_X and SHX_H and the average degree decreased. The centrality of *Proteobacteria* and *Chloroflexi* became higher in rhizobacteria of SHX_H than in those of SHX_S, while the centrality of *Planctomycetes* and *Actinobacteria* was higher in rhizobacteria of SHX_X. SD_S and SD_D had the same number of nodes (35), and the numbers of the edges of SD_D were higher than those of SD_S, indicating the more complicated interaction network of SD_D (Fig. 7c and Table S3).

**(iii) Relationships between soil physicochemical properties and rhizosphere soil bacteria.** Figure 8a depicts the correlations between rhizosphere soil properties

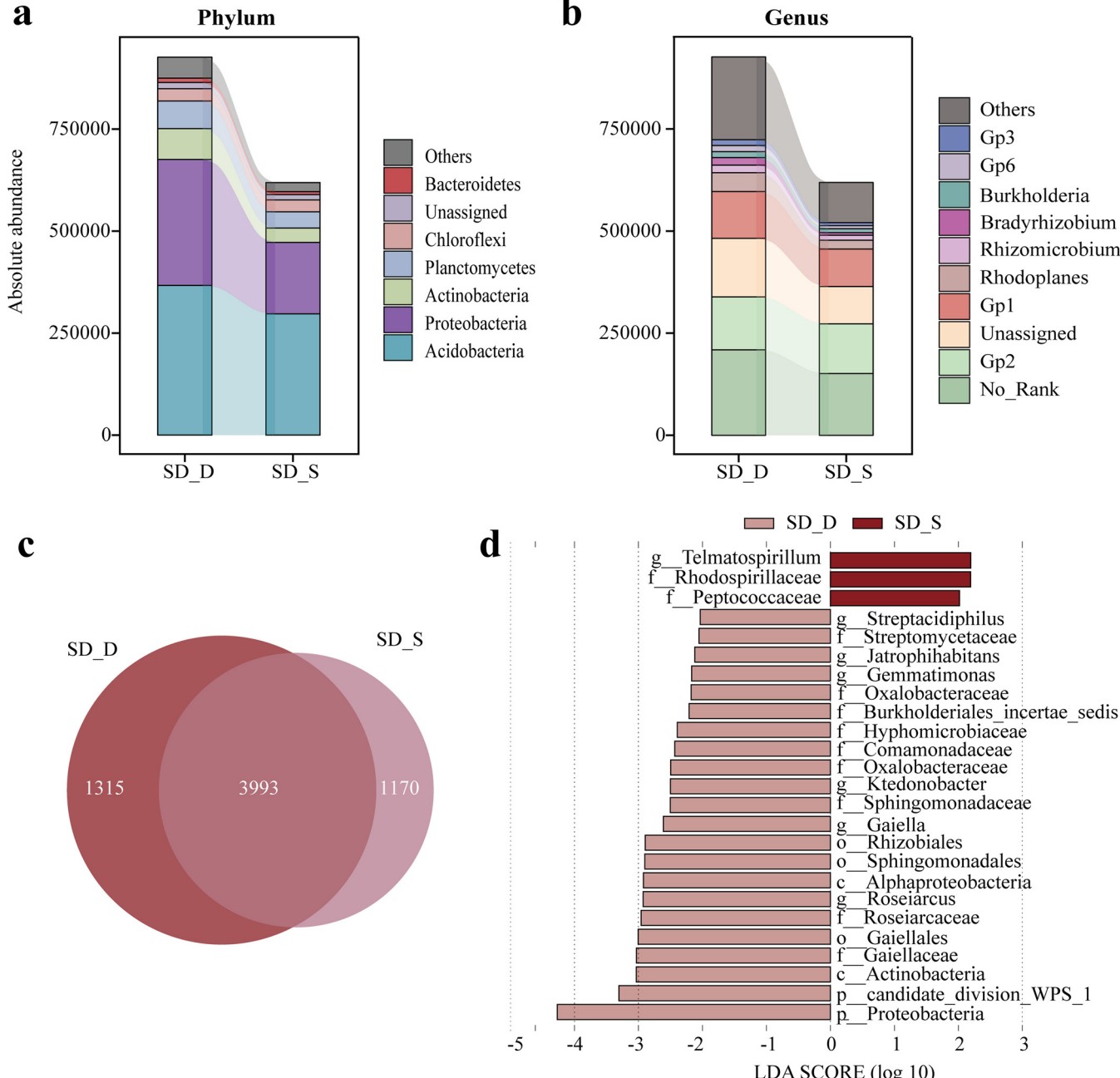

**FIG 5** Rhizosphere bacterial community composition and bacteria with significant differences in different tree species at SD. (a) Rhizosphere bacterial community composition at the phylum level. (b) Rhizosphere bacterial community composition at the genus level. (c) Bacterial OTU distribution in rhizosphere soil. (d) LDA scores of taxa enriched in different tree species at SD. Only taxa with LDA values of >2.2 (*P* < 0.05) are shown.

and the absolute abundance of rhizosphere bacteria of different tree species in SHX at the phylum level. The pH had a significant positive correlation with *Nitrospirae* and *Latescibacteria* (*P* < 0.05), as did SOM with *Bacteroidetes*. There was a significant positive correlation between AN and *Verrucomicrobia* or "*Candidatus* Omnitrophica" (*P* < 0.01). Furthermore, AP was positively correlated with *Proteobacteria*, *Planctomycetes*, and *Acidobacteria* (at a *P* value of <0.05) and strongly so with *Actinobacteria* and *Bacteroidetes* (*P* < 0.01). According to the RDA (Fig. 8c), soil pH, SOM, SOC, TN, TP, AN, and AP all contributed to changes in the rhizosphere bacterial community. Both pH and AP significantly influenced the rhizosphere bacterial community of different tree species in SHX.

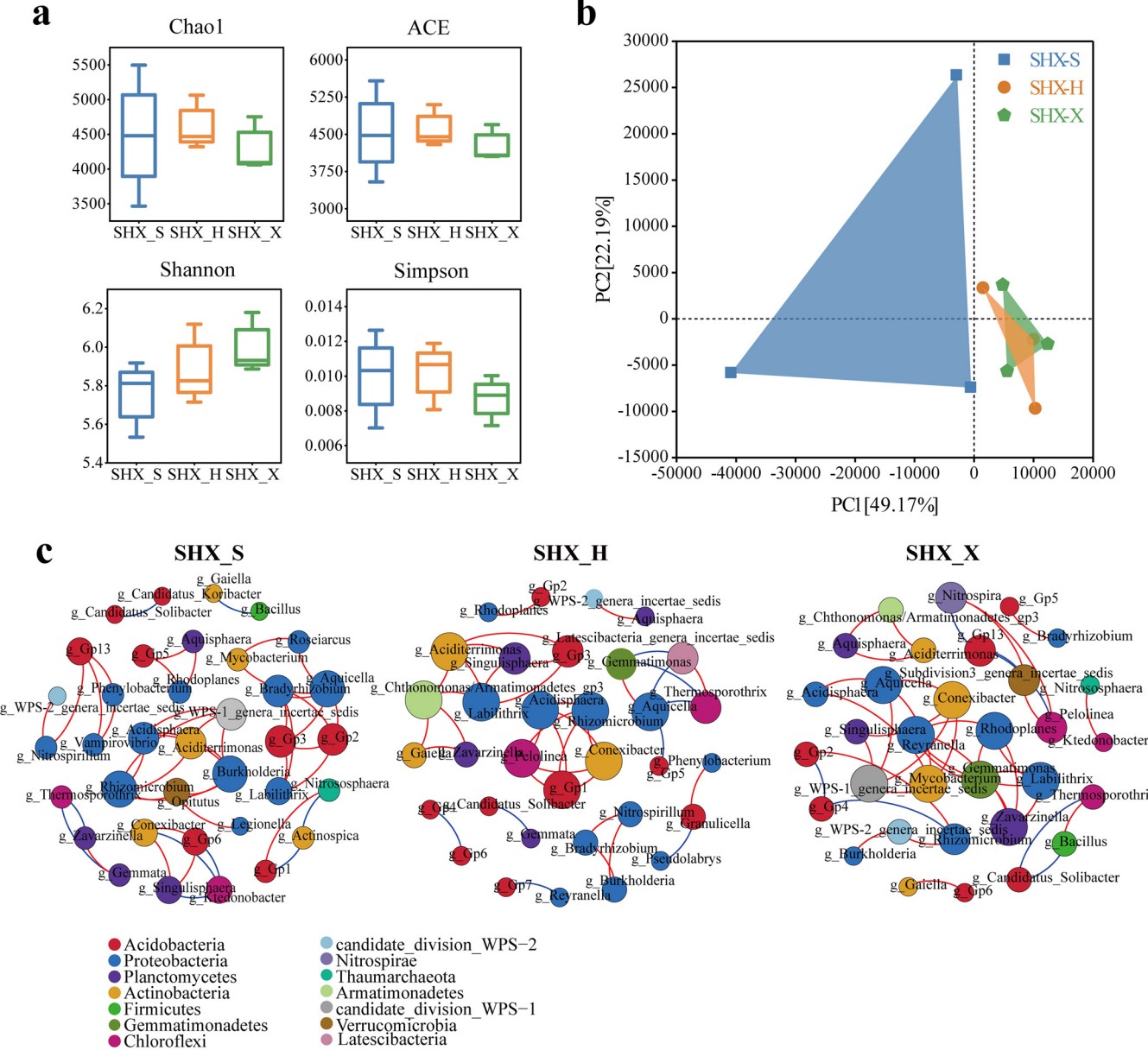

**FIG 6** Bacterial diversity and bacterial interaction networks in different tree species at SHX. (a) Alpha diversity of rhizosphere bacterial communities. (b) Principal-component analysis (PCA) of bacterial communities in the rhizosphere soil. (c) Interaction network of dominant members of the microbiota at the genus level (top 40) in rhizosphere soil of SHX_S, SHX_H, and SHX_X. The different colors indicate the corresponding taxonomic assignments at the phylum level. The edge colors represent positive (red) and negative (blue) correlations. Only significant interactions are shown ($r > 0.6$; $P < 0.05$).

In SD, Except for BRC1, no significant correlation was found for rhizosphere soil bacteria vis-à-vis the soil properties (Fig. 8b). However, TP featured a robust, positive correlation with *Nitrospirae*, and soil organic matter was markedly correlated with *Bacteroidetes* and *Gemmatimonadetes*. Rhizosphere soil nutrients that affected the rhizosphere bacterial communities of SD_S and SD_D in SD were mainly pH, SOM, TN, TP, and AN (Fig. 8d). The dominant bacteria phyla had positive correlations with SOM, TN, TP, and AN but had negative correlations with pH.

## DISCUSSION

**Effects of near-natural forest management and tree species on rhizosphere soil physicochemical properties.** Soil nutrients, particularly those available to plants, are intimately related to plant productivity and thus are often used as indicators of soil

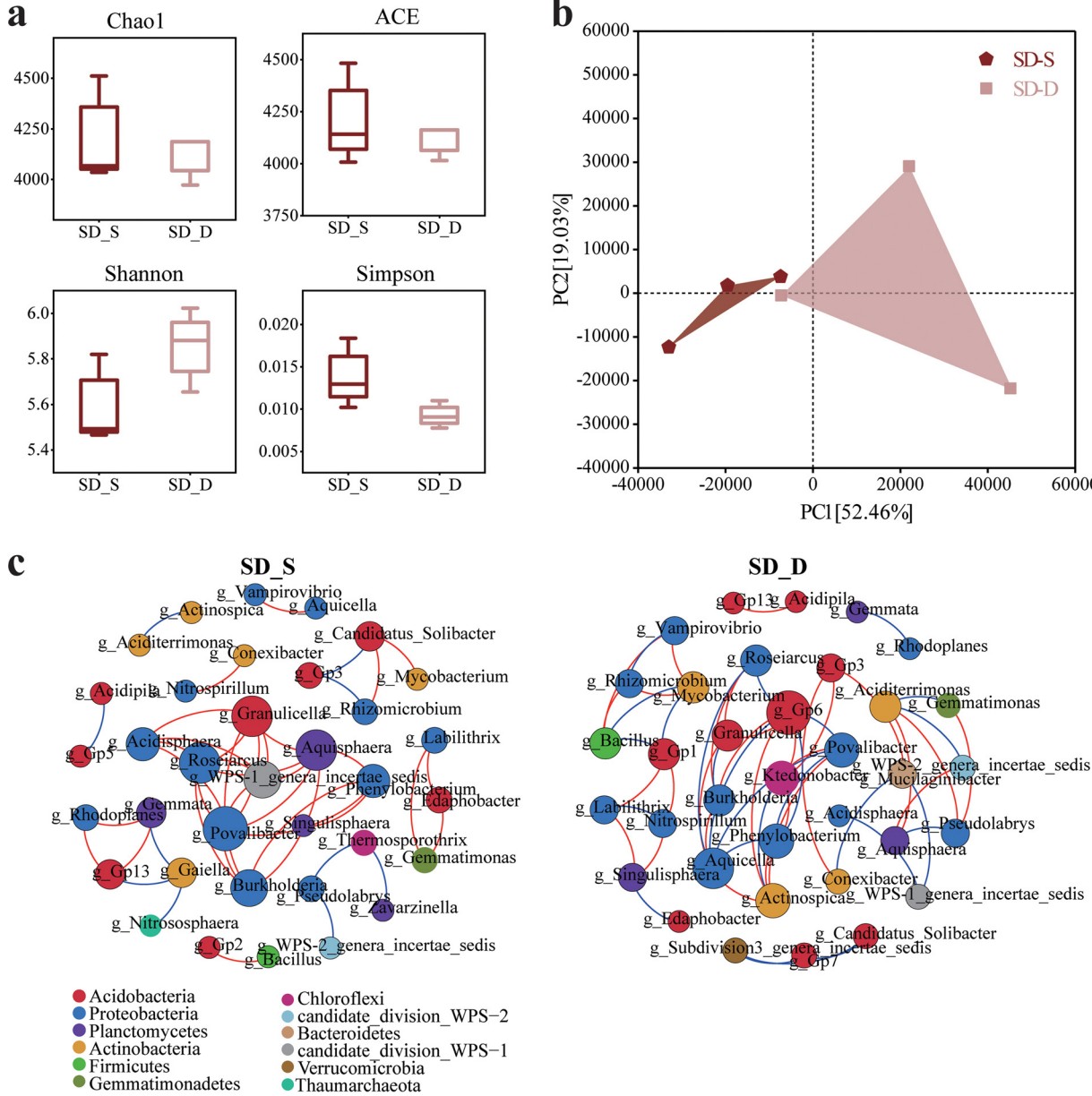

**FIG 7** Bacterial diversity and bacterial interaction networks in different tree species at SD. (a) Alpha diversity of rhizosphere bacterial communities. (b) Principal-component analysis (PCA) of bacterial communities in the rhizosphere soil. (c) Interaction network of dominant microbiota at the genus level (top 40) in rhizosphere soil of SD_S and SD_D. The different colors indicate the corresponding taxonomic assignments at the phylum level. The edge colors represent positive (red) and negative (blue) correlations. Only significant interactions are shown ($r > 0.6$; $P < 0.05$).

fertility quality (47). According to our study's results, the introduction of broad-leaved tree species into pure Chinese fir forest plantations enhanced the soil pH and the organic matter, organic carbon, total nitrogen, and alkali-hydrolyzable nitrogen contents in the rhizosphere soil of Chinese fir trees (Fig. 1). These findings are similar to those reported previously by Q. K. Wang and S. L. Wang (48), although the differences in some nutrient indicators were not significant. The total phosphorus (TP) contents of pure Chinese fir stands declined sharply compared with those of the mixed stand types, indicating that phosphorus may be a limiting factor for the growth of Chinese fir. Recently, L. Zhou et al. (49) found that the SOM, AN, and phosphorus contents of pure Chinese fir forests in southeastern China were significantly lower than those of mixed forests. Previous studies have reported that the introduction of broad-leaved tree species can improve the physical and chemical properties of soil (50, 51); however,

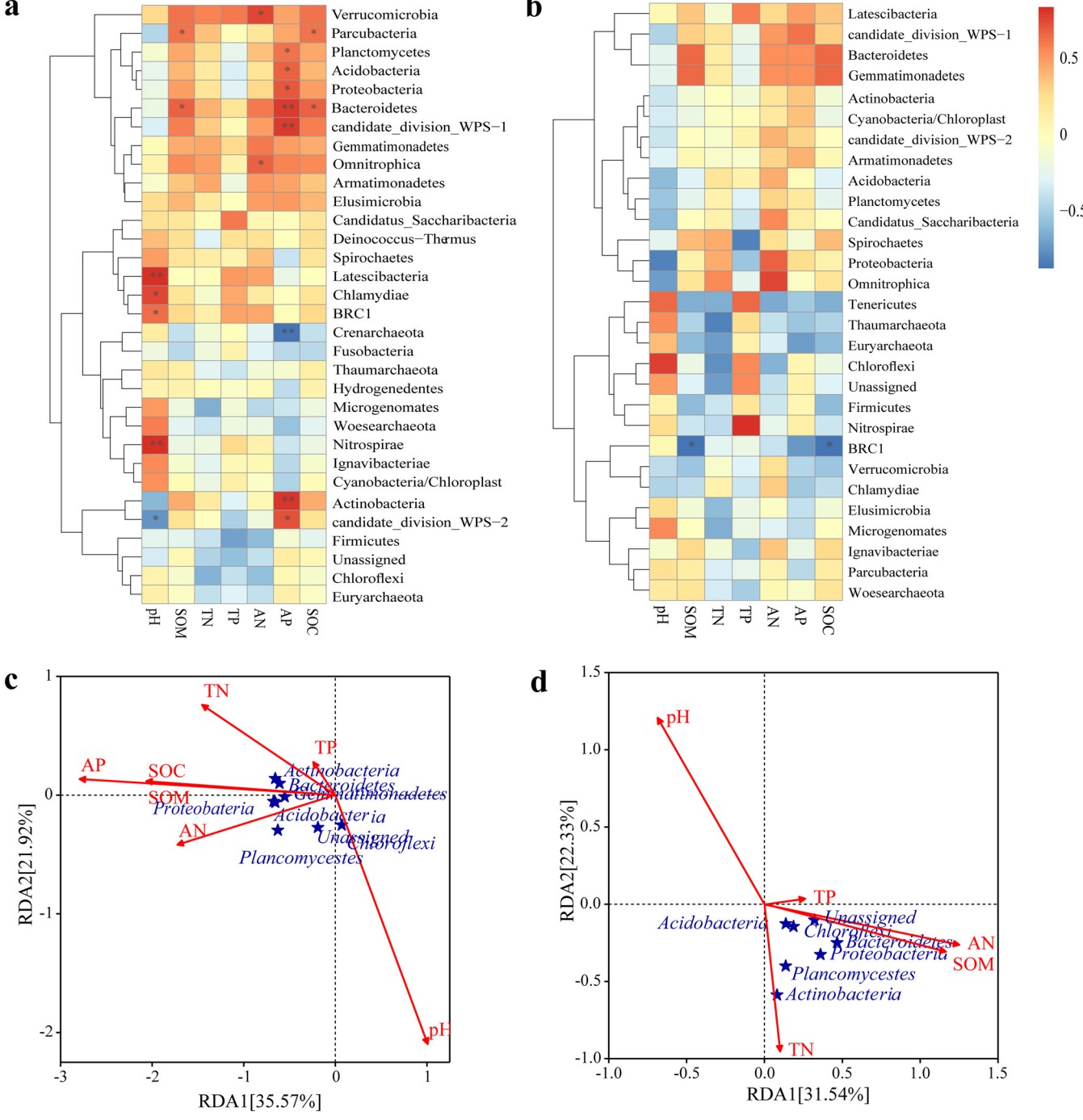

**FIG 8** Correlation relationships (SHX [a] and SD [b]) and redundancy analysis (RDA) (SHX [c] and SD [d]) between rhizosphere bacteria and rhizosphere soil physicochemical properties in different tree species of SHX and SD at the phylum level. *, $P < 0.05$; **, $P < 0.01$.

the introduction of various tree species often has differential effects on the degree of improvement (52), which could be due to the distinctive composition and decomposition characteristics of their litters in different mixed forests (53). More litter tends to increase the SOM content, nitrogen content, microbial community, and enzyme activity in Chinese fir plantations (54). There are usually differences in the soil chemical properties between pure forests and mixed forests, but the latter does not always have a better soil nutrient status than the former (55, 56).

Among different tree species, the physical and chemical properties of their rhizosphere soil differed little. The AP concentration in rhizosphere soil was significantly higher for

Chinese fir than for either *Castanopsis hystrix* or *M. hedyosperma* in SHX. The values of other nutrients revealed that Chinese fir > *Castanopsis hystrix* > *Michelia hedyosperma* (Fig. 4a). In SD, except for the pH and the TP content, the contents of SOM, TN, AN, and AP in the rhizosphere of Chinese fir were lower than those of *Castanopsis fissa*. The rhizosphere soil of *Castanopsis fissa* had a pH of 4.47, which was lower than that of Chinese fir (Fig. 4b). The availability of phosphorus is one of the limiting factors for microbial growth (57). Different root exudates of different tree species, as well as the release of rhizosphere exudates, can promote the transformation of soil P from a difficult-to-use form into an easy-to-use form (58, 59), and rhizosphere microorganisms may increase the availability of phosphorus by releasing phosphatase (60).

**Effects of mixed patterns on the structure and diversity of bacteria in the rhizosphere soil of Chinese fir.** Generally, the dominant phyla of Chinese fir soil bacterial communities are considered to be *Acidobacteria*, *Proteobacteria*, and *Actinomyces* (61, 62). We found, however, that in pure Chinese fir and two mixed forest stand types, the dominant phyla were *Proteobacteria* and *Acidobacteria*. The phylum *Acidobacteria* is composed mainly of acidophilic bacteria, and among soil properties, pH has the strongest correlation with acidophilic bacteria (63, 64). Tree species, followed by soil properties, influenced the composition and diversity of bacterial communities in forest soil (65–67). The long-term pure-plantation model aggravated the microecological imbalance in the rhizosphere soil of Chinese fir and significantly reduced the diversity and metabolic activity of the soil microbial community (68). The introduction of different tree species had different effects on the bacterial community in the rhizosphere of Chinese fir. The total absolute abundances of Chinese fir rhizosphere bacteria in SHX and SD increased in comparison with those in a pure Chinese fir forest (S), and their OTUs also likewise increased albeit to various degrees (Fig. 3). In SHX, the Chinese fir rhizosphere bacterial community had more OTUs and greater bacterial diversity, but in SD, it had the highest absolute abundance.

The absolute abundances of *Planctomycetes* and *Actinobacteria* were significantly higher in the rhizosphere soil of SD_S than in that of S_S, while the absolute abundances of *Verrucomicrobia* and *Actinobacteria* were significantly higher in the rhizosphere soil of SHX_S than in that of S_S (see Fig. S2 and S3 in the supplemental material). The LEFSE results corroborated the ANOVA results (Fig. 1d): *Actinobacteria* were significantly enriched in SHX_S, while *Planctomycetaceae* were significantly enriched in SD_S. This suggested that they may play important roles in the near-natural cultivation of Chinese fir and may be the bacteria that play different roles in different mixed patterns. Genera of the *Planctomycetaceae* have the potential for the production of exopolysaccharides and lipopolysaccharide, which is beneficial for the formation of microaggregates in soil (69). *Actinobacteria* mainly promote the soil decay of animal and plant remains. Some reported *Actinobacteria* can produce related enzymes for decomposing lignin and cellulose (70). For example, *Thermobifida* was found to efficiently bind extracellular enzymes with xylanase (71). *Opitutus*, belonging to the *Verrucomicrobia*, is known as a specialized anaerobic bacterium and was previously associated with nitrogen fixation in rice fields under strictly anaerobic conditions (72).

Abundances determine the functional roles of bacteria in complex communities, and microbial communities featuring greater diversity are thought to be more resistant to pathogen invasion and more stable (73, 74). Our results showed that the introduction of *Michelia hedyosperma* and *Castanopsis hystrix* influenced the bacterial community in the rhizosphere of Chinese fir. This is similar to the findings of two other studies (75, 76) where the introduction of broad-leaved tree species into pure plantations changed the composition and structure of the soil microbial community. Studies have reported that the introduction of broad-leaved tree species in pure Chinese fir plantations improves the soil quality by increasing organic matter, effective nutrients, and soil microbial activity (48, 77). This may be because after their introduction, the average amount of annual litter increased considerably, and its composition changed, which hastened the input of nutrients from the ground surface (78, 79) and indirectly led to changed contents of nutrients in the soil.

Microbial network analysis, by identifying the species with high connectivity

throughout the network or the position located within the species module, allows the identification of the key species and the more important species in the entire network, and these species may have a deterministic role in the structure and function of the microbial community (80). The topological characteristics of the cooccurrence network showed that the community complexity of rhizosphere bacteria of SD_S and SHX_S was increased (Fig. 2c; Table S1). The key nodes belonged mainly to *Acidobacteria*, *Proteobacteria*, *Planctomycetes*, and *Actinobacteria*, which may perform key ecological functions associated with near-natural cultivation.

The diversity of tree species indirectly affects soil microbial diversity (81), and the greater bacterial diversity in the rhizosphere of SHX_S may have arisen from the greater diversity of its aboveground tree species and the more complex composition of the litter, which indirectly impacts rhizosphere bacterial diversity. Furthermore, according to the type and quantity of vegetation litter, in addition to root exudates, especially their chemical characteristics, it can selectively stimulate the growth of soil microorganisms, thereby influencing the characteristics of the microbial community (82, 83).

**Effects of tree species on the structure and diversity of bacteria in rhizosphere soils.** In the same mixed stand type, the rhizosphere bacterial communities of broad-leaved tree species and Chinese fir had similar compositions at the phylum and genus levels, similar to other reports (40, 84, 85), although there were stark differences in their absolute abundances. In SHX, the absolute abundance of rhizosphere bacteria declined successively across Chinese fir, *Michelia hedyosperma*, and *Castanopsis hystrix*, and this difference was reflected mainly in *Actinobacteria* (Fig. 4d and Fig. S2). The results of the network analysis showed that the diversity and complexity of rhizosphere bacterial communities of Chinese fir were greater than those of *Castanopsis hystrix* and *Michelia hedyosperma* (Fig. 6b). The rhizosphere bacterial richness was higher under *Castanopsis hystrix* than under Chinese fir, being the lowest under *Michelia hedyosperma*. Conversely, the Shannon index for rhizosphere bacteria of *Michelia hedyosperma* was the highest, revealing that this species supported the highest level of bacterial community diversity. In SD, the absolute abundance of bacteria was lower under Chinese fir than under *Castanopsis fissa*, but this trend was reversed for their Shannon indices. *Proteobacteria* were enriched in rhizobacteria of SD_D, and its rhizosphere bacterial community complexity was higher than that of SD_S (Fig. 5d and Fig. S3). According to the results shown in Fig. 5d, the absolute abundances of bacteria related to nitrogen fixation, such as *Burkholderiales* and *Rhizobiales*, were higher in SD_D than in SD_S (86, 87). The *Sphingomonadaceae* is a member of hydrocarbon degradation bacteria and is known for its ability to degrade a variety of polycyclic aromatic hydrocarbons (88, 89). Our results showed that different rhizosphere bacteria were recruited from the roots of different tree species, which may be due to the different root exudates of the different tree species (90, 91). Plants transport photochemical products to the rhizosphere and form rhizosphere sediments, which promote the growth, metabolism, and accumulation of soil microorganisms in the rhizosphere (92). The different abilities of roots to transport carbon sources to the rhizosphere soil also lead to the selective shaping of the rhizosphere soil microbial species and their community structure (93, 94). Small-molecule organic substances in Chinese fir root exudates consist mostly of carbohydrates, followed by organic acids and then amino acids, ranking third (95). There are pronounced variations in the types and quantities of various low-molecular-weight sugars, electrodeless ions, and organic acids in the root exudates of different tree species (96). In this respect, the flavonoids, phenolic compounds, citric acid, and malic acid in the root exudates are quite different (97, 98). All of these discrepancies can affect the final composition of the rhizosphere soil bacterial community. The composition of root exudates depends on the plant species, and this dependence was thought to affect the microbial community in the rhizosphere soil (16, 99, 100). It had been shown that the root exudates of specific plant species cultivated their own soil fungal communities (101). In addition, the growth metabolism and litter matrix quality vary among different plants' roots, root metabolism, and litter decomposition, which will jointly alter the soil properties (especially those of rhizosphere soil) to determine the structure and diversity of bacterial communities in rhizosphere soils of different plants (102, 103).

**Effects of soil properties on soil bacterial communities.** The vegetation community is a crucial source of organic nutrients and energy for soil bacteria to survive. Vegetation can affect soil bacterial community diversity by changing the physical and chemical properties of the soil beneath it, for example, its pH and nutrient content (104). Many studies have shown that the soil bacterial community structure is closely related to soil properties (27, 85, 105). In our study, within different treatments (stand types and tree species), the soil physiochemical factors that exerted important effects on the rhizosphere bacterial community were not the same. For Chinese fir in mixed stands, SOM and TN were the important variables shaping the rhizosphere bacterial community, but for different species in the same type of mixed forest, the limiting factors were changed, mainly to pH. Soil pH, alkali-hydrolyzable nitrogen, and available phosphorus were the key influencing factors driving changes in soil bacterial community structures (85, 105, 106). Like many other previously reported findings, we found that rhizosphere soil pH was the main factor affecting the community structure of rhizosphere soil bacteria in Chinese fir plantations (107). Soil pH is an important determinant of bacterial diversity and community structure on a global scale, and soil bacterial diversity generally peaks in neutral soils and is lower in acidic soils (108). Some studies found more available phosphorus in the rhizosphere soil of Chinese fir mixed forests than in pure Chinese fir forests (109, 110), contrary to our results here. Yet the formation of mixed forests tends to reduce the content of available phosphorus on a global scale (111). It is worth noting that unlike in SD, the content of AP in the rhizosphere soil of Chinese fir in SHX was significantly different from those of the two broad-leaved species, perhaps because phosphate-solubilizing microorganisms (PSMs) were present. PSMs are a kind of microorganism that can convert forms of phosphorus difficult to take up by plants into a more usable state readily absorbed by plants (112, 113). Currently, the reported PSMs include mainly *Pseudomonas*, *Bacillus*, *Rhizobium*, *Burkholderia*, *Achromobacter*, *Agrobacterium*, *Micrococcus*, *Aerobacter*, *Flavobacterium*, and *Erwinia*, etc. (114); most of these bacteria belong to the *Proteobacteria*. Accordingly, as inferred from the results shown in Fig. 5a, *Proteobacteria* were more abundant in SHX_S than in the two broad-leaved species (SHX_X and SHX_H), making them the most likely explanation for the differences in AP contents found.

**Conclusion.** Our study demonstrated that forest conversion caused by implementing near-natural forest management practices is a critical factor that impacts the rhizosphere soil properties and bacterial composition of Chinese fir plantations. The rhizosphere soil qualities of Chinese fir were ameliorated after conversion from pure to mixed forest stands. Notably, the content of phosphorus was significantly improved. The structure and diversity of the soil bacterial community did not change markedly, but the conversion nonetheless led to increases in not only the number of soil bacterial OTUs but also their absolute abundance. Not only that, the introduction of broad-leaved tree species increased the abundance of beneficial bacteria in the soil, such as *Planctomycetes* and *Actinobacteria*, etc. At the same time, tree species identity was also a contributing factor affecting the rhizosphere's soil properties and bacterial community. In the *Cunninghamia lanceolata*-*Castanopsis hystrix*-*Michelia hedyosperma* mixed plantation, both the organic nutrient content and the absolute abundance of bacteria under Chinese fir were higher than those for cooccurring broad-leaved species. On the contrary, these indicators were lower for Chinese fir than for *Castanopsis fissa* in the *Cunninghamia lanceolata*-*Castanopsis fissa* mixed plantation. *Acidobacteria* and *Proteobacteria* were chiefly responsible for the different compositions of soil bacteria. Moreover, changes in soil pH and soil organic matter also significantly affected soil bacterial communities, indicating that both factors contributed to rhizosphere soil bacterial compositional shifts associated with near-natural forest management and tree species. Above all, the stand formation of two mixed types of forest is capable of greatly impacting the rhizosphere soil and its bacterial communities. Accordingly, the further introduction of *Michelia hedyosperma* and *Castanopsis hystrix* may be more conducive to improving soil quality and maintaining the long-term productivity of Chinese fir plantations.

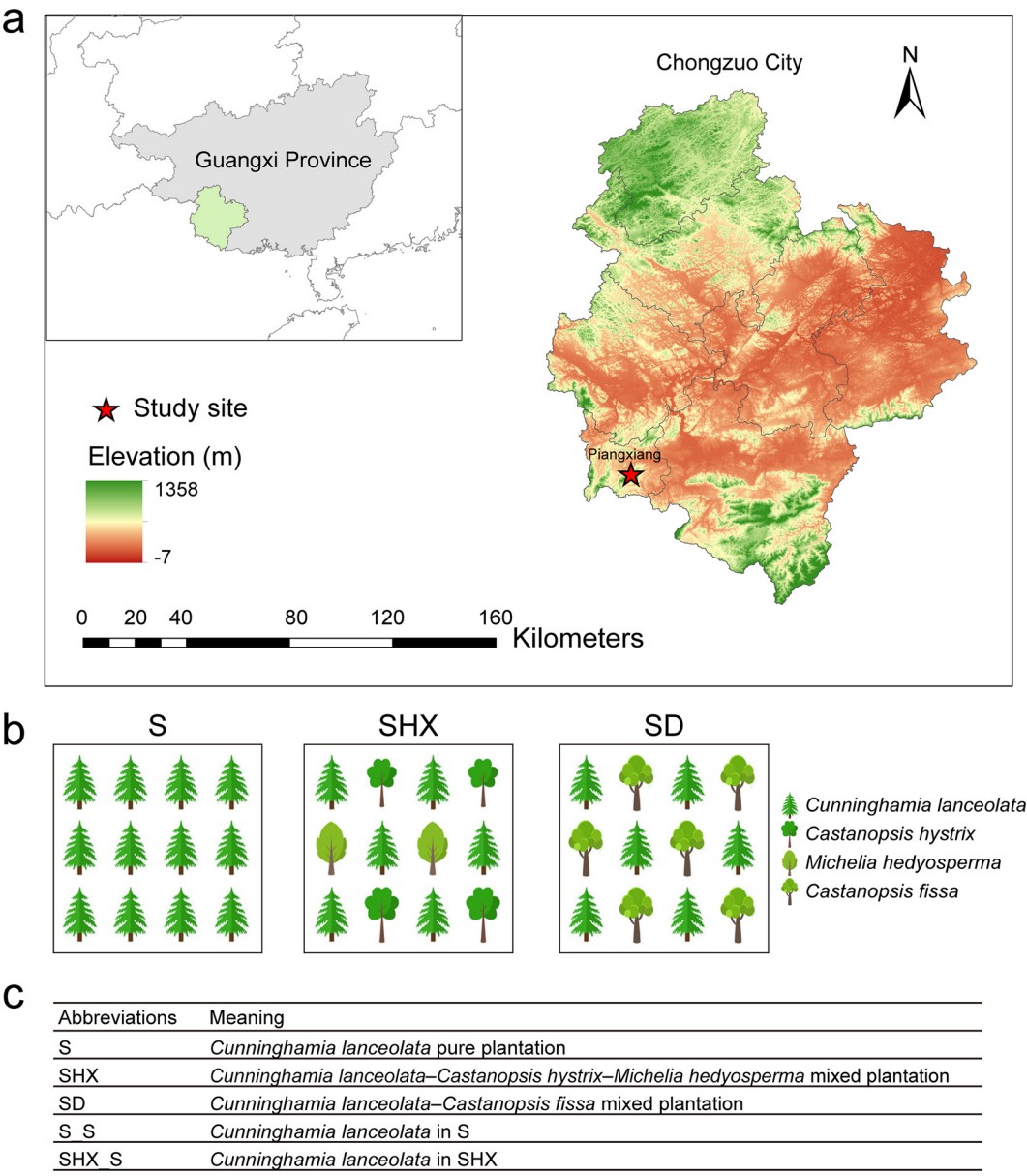

**FIG 9** (a and b) Study area (a) and sampling site (b) in Pingxiang, Guangxi Province, People's Republic of China. (c) Abbreviations.

| Abbreviations | Meaning |
|---|---|
| S | *Cunninghamia lanceolata* pure plantation |
| SHX | *Cunninghamia lanceolata–Castanopsis hystrix–Michelia hedyosperma* mixed plantation |
| SD | *Cunninghamia lanceolata–Castanopsis fissa* mixed plantation |
| S_S | *Cunninghamia lanceolata* in S |
| SHX_S | *Cunninghamia lanceolata* in SHX |
| SHX_H | *Castanopsis hystrix* in SHX |
| SHX_X | *Michelia hedyosperma* in SHX |
| SD_S | *Cunninghamia lanceolata* in SD |
| SD_D | *Castanopsis fissa* in SD |

## MATERIALS AND METHODS

**Study site.** The experimental site was situated on the Fubo Forest Farm (106°51′E, 22°03′N) in Daqingshan, Pingxiang City, Guangxi Province, People's Republic of China, located in the low-latitude area (South Subtropical Area) (Fig. 9a). With an annual average temperature of 19.5°C to 21.41°C and average annual precipitation of 1,376 mm, it has a subtropical monsoon climate. The main geomorphic types in this area are low mountains and hills, with latosol and red soil types predominating. Since 1993, the near-natural experimental forest has been afforested with an initial density of 3,000 plants·ha$^{-1}$, followed by forest management that includes two rounds of stand thinning, for a final retention density of 375 to 585 plants·ha$^{-1}$. In 2008, saplings of broad-leaved trees were planted under the canopy of Chinese fir, namely, *Michelia hedyosperma* Y. W. Law, *Castanopsis hystrix* Miq., *Castanopsis fissa* (Champ. ex Benth.) Rehder & E. H. Wilson, and *Erythrophleum fordii* Oliv. Finally, two mixed plantations were formed: a *Cunninghamia lanceolata-Castanopsis hystrix-Michelia hedyosperma* mixed plantation (SHX) and a *Cunninghamia lanceolata-Castanopsis fissa* mixed plantation (SD). The different mixed stands and their abbreviations are shown in Fig. 9b and c.

In 2019, both mixed plantations (SHX and SD) along with a pure Chinese fir stand (S) were selected as research objects. A sample circle with a 10-m radius was centered on a well-growing Chinese fir tree (disease-free) and one (in SD) or two (in SHX) broad-leaved mixed tree species closest to it as a single sampling unit. Each plantation forest type was set with three sample circles as replicates. The pure Chinese fir forest was set with 3 sample circles of the same size centered on average wood. Hence, in this way, a total of 9 sample circles were set under basically the same site conditions.

**Soil sampling.** Surface weeds and loose soil were removed from the base of the tree trunk, and the rhizosphere soil was collected by using the shake-off method in a 0- to 20-cm soil layer, 50 cm from the base of the trunk (115). For the same sample circle, we equally mixed the rhizosphere soil of the same tree species. These composite soil samples were then sieved (2-mm mesh), and each sample was divided into two parts: one portion was stored at −80℃ for microbial high-throughput sequencing, and the other part was stored at 4℃ and used for the determination of soil physicochemical properties.

**Soil physicochemical analysis.** Soil organic matter (SOM) was quantified using the K dichromate external heating method. The concentrations of total N (TN) and hydrolytic N (AN) were determined by the Kjeldahl method and the diffusion absorption method, respectively (116). The total P (TP) concentration was measured by NaOH alkali dissolution–Mo-Sb colorimetry, while the available P (AP) concentration was determined by NaHCO$_3$ alkali dissolution–Mo-Sb anticolorimetry. The soil pH was measured by the potentiometer method (water-soil, 2.5:1). For applying these determination methods, we followed the instructions in *Soil Agricultural Chemical Analysis* (117).

**Soil bacterial community analysis.** The absolute quantification of 16S rRNA genes was performed externally by Genesky Biotechnologies Inc. (Shanghai, People's Republic of China). The PowerSoil DNA isolation kit (MoBio, Carlsbad, CA, USA) was used to extract total genomic DNA. Multiple spike-ins with identical conserved regions for 16S rRNA genes and variable regions replaced by random sequences with ~40% GC content were artificially synthesized, and an appropriate mixture with known gradient copy numbers of spike-ins was then added to the sample DNA. A MiSeq sequencer was used to amplify and sequence the V4-V5 region and spike-ins of the 16S rRNA gene to generate at least 10 million 2× 250-bp paired-end raw reads (Genesky) (118, 119).

Next, TrimGalore (v0.4.5; Babraham Bioinformatics, UK) and Mothur (v1.39.3 [https://www.mothur.org/]) were used to remove the adaptor and primer sequences, respectively. At the opposite end, the reads were merged and filtered, and the remaining reads were then clustered into operational taxonomic units (OTUs) with a sequence similarity level of at least 97%. The OTUs were annotated, and a standard curve was established between the read count and the number of incorporated DNA copies. The absolute copy number of the bacterial OTUs can be calculated by using the read count of the corresponding bacterial OTUs.

**Statistical analyses.** Graphical representations were generated with OriginPro version 2021 (OriginLab Corporation, Northampton, MA, USA). One-way analysis of variance (ANOVA), followed by a Duncan multiple-comparison test, was used to test for differences in the physical and chemical properties of the rhizosphere soils among the three plantation stand types and tree species. Bacterial alpha diversity, including the Chao1 index, the ACE index, the Shannon index, and the Simpson index, was calculated using Mothur (v1.39.3). Pearson correlations were tested between rhizosphere soil physicochemical properties and the absolute abundances of the bacterial community, with redundancy analysis (RDA) being used to explain the effects of the rhizosphere soil physicochemical properties on the bacterial community species composition. Principal-component analysis (PCA) was used to convey the similarities and differences in the compositions of microbial communities among different tree species and different mixed forests. SPSS (v22.0; IBM, Armonk, NY, USA) was used to carry out the ANOVAs, while R software (v3.4.3) was used for the correlation analysis, RDA, and PCA. A bacterial interaction network (using Pearson correlations) was performed using the OmicStudio tools (https://www.omicstudio.cn/tool), and the network map was made using Gephi 0.9.4. The LDA effect size (LEFSE) was calculated and plotted using Galaxy (https://huttenhower.sph.harvard.edu/galaxy/).

**Data availability.** The 16S rRNA gene sequences used in this study have been submitted to the NCBI Sequence Read Archive (SRA) under BioProject accession number PRJNA847326.

## SUPPLEMENTAL MATERIAL

Supplemental material is available online only.

**SUPPLEMENTAL FILE 1**, PDF file, 0.5 MB.

## ACKNOWLEDGMENTS

We are grateful to the technical staff of the Experimental Center of Tropical Forestry, Chinese Academy of Forestry, Pingxiang City, Guangxi Province, People's Republic of China, for sampling and providing useful suggestions, and we appreciate the technical support provided by Genesky Biotechnologies Inc. (Shanghai, People's Republic of China).

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
