## [Reviewer comments · Microbiology Spectrum]

Microbiology Spectrum

Response of Rhizosphere Bacterial Communities to Near Natural Forest Management and Tree Species Within Chinese Fir Plantations

Jie Lei, Hanbin Wu, Xiaoyan Li, Wenfu Guo, Aiguo Duan, and Jianguo Zhang

Corresponding Author(s): Aiguo Duan, Chinese Academy of Forestry Research Institute of Forestry

Review Timeline:

Submission Date:	June 21, 2022
Editorial Decision:	October 7, 2022
Revision Received:	October 21, 2022
Editorial Decision:	November 30, 2022
Revision Received:	December 2, 2022
Accepted:	January 3, 2023

Editor: Kevin Theis

Reviewer(s): Disclosure of reviewer identity is with reference to reviewer comments included in decision letter(s). The following individuals involved in review of your submission have agreed to reveal their identity: Modupe Stella Ayilara (Reviewer #3)

Transaction Report:

DOI: <https://doi.org/10.1128/spectrum.02328-22>

October 7, 2022

Prof. Aiguo Duan
Chinese Academy of Forestry
Beijing
China

Re: Spectrum02328-22 (Near-Natural Forest Management and Tree Species Affect the Chinese fir plantation Rhizosphere Soil Properties and Rhizosphere Bacterial Communities)

Dear Prof. Aiguo Duan:

Thank you for submitting your manuscript to Microbiology Spectrum. I sincerely apologize for the delayed decision. The manuscript has now been reviewed by two experts in the discipline. You will see below that some concerns were noted, yet they are addressable.

Link Not Available

Sincerely,

Kevin R. Theis

Journals Department
Reviewer comments:

Reviewer #2 (Comments for the Author):

The manuscript aims to evaluate the effect of near-natural forest management on rhizosphere bacterial communities in Chinese fir plantations. The general approach of the study was to compare the bacterial and soil biochemistry profiles of a pure Chinese fir (*Cunninghamia lanceolata*) plantation (S), a *Cunninghamia lanceolata*-*Castanopsis hystrix*-*Michelia hedyosperma* mixed plantation (SHX), and a *Cunninghamia lanceolata*-*Castanopsis fissa* mixed plantation (SD). This is a descriptive study.

The research question is clear, the conclusions are supported by the results, and the results will be of value to the broader research community.

The concerns/suggestions that I do have largely pertain to presentation and clarification.

1) I strongly recommend a graphical abstract or diagram to clearly indicate the symbols/abbreviations used for the forest types and trees. It is very difficult to keep track of this, especially as the abbreviations are not acronyms.

2) In Lines 249-251, it is not clear what absolute abundance here means. If it means total number of sequences obtained per sample per forest, then that is a technical issue rather than a biological one. It is typical to conduct equimolar pooling of multiplexed samples prior to loading the library onto the MiSeq instrument. If that was done, it here did not work. It is one thing to talk about the unique approach to the "absolute abundance" of individual bacterial taxa (when the total DNA from each library is equal), yet another entirely to speak of "absolute abundance" of soil samples in general. A key potential technical explanation for observed differences are that different forest soils have different amounts of PCR inhibitors. There can also be processing issues if the samples were not processed in randomized fashion. [This same issue arises in the tree-specific analyses later in the manuscript. Just a couple of sentences.]

Minor comments:

Methods

- In Microbiology Spectrum, the Methods come after the Discussion section.

Results

- Line 405: There are several places in the manuscript wherein "OTU" is misspelled as "OUT"

- In general, I recommend not reporting in the text the:

- 1) percent variance explained by PCA axes; this information is in the figures
- 2) specific numbers of nodes and edges in networks for specific forests and trees
- 3) etc.

The results are summarized in the text and the specifics are in the figures. If a reader desires to focus on them, they can consult the figures. At present, the Results section is a very difficult read. More narrative reporting and less specific recounting of results that are also presented in figures and tables would make the manuscript easier to read and digest.

- Throughout the manuscript, is Simpson being reported as the Simpson Index, or is it being reported as 1- Simpson or Inverse Simpson? Please be explicit about this in the Methods.

Reviewer #3 (Comments for the Author):

The manuscript discusses the effect of Near-natural forest management and tree species on Chinese fir plantation's rhizosphere soil and bacterial communities. English editing is recommended by a native English language speaker
Considering rephrasing the topic for clarity "Effect of Near-natural Forest Management and Tree Species on Chinese Fir Plantation Rhizosphere and Bacterial Communities"

L15-The font size for the word "Abstract" is not consistent

L26- remove the word "did"

L26 and L27- replace the word "enriched" with "abundant"

L30- replace nitrogen fixation-related with "nitrogen fixing"

L31-replace "present" with "abundant"

L35-L38- rephrase for clarity"

Other corrections are indicated in the comment section of the pdf

Staff Comments:

Preparing Revision Guidelines

- Point-by-point responses to the issues raised by the reviewers in a file named "Response to Reviewers," NOT IN YOUR

COVER LETTER.

- Upload a compare copy of the manuscript (without figures) as a "Marked-Up Manuscript" file.
- Each figure must be uploaded as a separate file, and any multipanel figures must be assembled into one file.
- Manuscript: A .DOC version of the revised manuscript
- Figures: Editable, high-resolution, individual figure files are required at revision, TIFF or EPS files are preferred

Please return the manuscript within 60 days; if you cannot complete the modification within this time period, please contact me. If you do not wish to modify the manuscript and prefer to submit it to another journal, please notify me of your decision immediately so that the manuscript may be formally withdrawn from consideration by Microbiology Spectrum.

**Near-Natural Forest Management and Tree Species Affect the Chinese fir**
**plantation Rhizosphere Soil Properties and Rhizosphere Bacterial Communities**

Jie Lei¹, Hanbin Wu¹, Xiaoyan Li¹, Wenfu Guo², Aiguo Duan^{1,3,*} and Jianguo Zhang^{1,3}

¹ State Key Laboratory of Tree Genetics and Breeding, Key Laboratory of Tree Breeding and
Cultivation of the State Forestry Administration, Research Institute of Forestry, Chinese Academy
of Forestry, Beijing 100091, PR China; duanag@caf.ac.cn (A.D.), leijiekyra@163.com (J.L.),
cafwuhanbin@163.com (H.W.), 3161566793@qq.com (X.L.), zhangjg@caf.ac.cn (J.Z.)

² Experimental Center of Tropical Forestry, Chinese Academy of Forestry, Pingxiang 532600,
China; 13878689910@139.com (W.G.)

³ Collaborative Innovation Center of Sustainable Forestry in Southern China, Nanjing Forestry
University, Nanjing 210037, PR China

* Correspondence: duanag@caf.ac.cn

**ABSTRACT**

Near-natural forest management plays an important role in maintaining the long-term
productivity and soil fertility of plantations. We conducted high-throughput absolute
quantitative sequencing of 16S rRNA to compare the structure and diversity of
rhizosphere soil bacterial communities among pure Chinese fir (*Cunninghamia*
*lanceolata*) plantation (S), *Cunninghamia lanceolata*—*Castanopsis hystrix*—*Michelia*
*hedyosperma* mixed plantation (SHX), and *Cunninghamia lanceolata*—*Castanopsis*
*fissa* mixed plantation (SD). The results revealed that near-natural forest management
improved the rhizosphere soil properties of Chinese fir, especially phosphorus content.
Rhizosphere soil bacterial communities of Chinese fir in SHX and SD contained
higher total absolute abundance and more unique operating classification units (OTUs)
than did the pure plantation forest. *Planctomycetes* and *Actinobacteria* were enriched
in SD, and *Actinobacteria* were enriched in SHX. Tree species also had an impact on
the rhizosphere soil bacterial communities. For the rhizosphere soil of different tree

species of SHX, AP content of the rhizosphere of Chinese fir significantly surpassed
that of *Castanopsis hystrix* and *Michelia hedyosperma*. Nitrogen fixation-related
bacteria such as *Burkholderiales* and *Rhizobiales* were more present in Chinese fir in
SD than *Castanopsis fissa*. *Actinobacteria* and *Proteobacteria* underpinned the
differences found in the composition of soil bacteria. The pH and soil organic matter
were key variables influencing the rhizosphere soil bacterial communities. Our results
showed that near-natural management of introduced broad-leaved tree species for 12
36 years can drive the alteration of physicochemical properties and bacterial community
structure and composition of rhizosphere soil in Chinese fir plantations, with tree
species identity further shaping the rhizosphere soil bacterial community.

**IMPORTANCE**

The near-natural forest management is an important way to change the soil
fertility decline and productivity reduction of pure Chinese fir plantations. At present,
many detailed studies have been carried out on the impact of near-natural forest
management on Chinese fir plantations at home and abroad. However, there are still
few studies on the response of rhizosphere bacterial communities to near-natural
forest management. Our study determined absolute quantification of Chinese fir
rhizosphere bacterial communities in different mixed patterns. The results underscore
the importance of near-natural forest management on Chinese fir plantation
rhizosphere bacterial communities, and provide new information on soil factors
affecting rhizosphere bacterial communities in South China.

**KEYWORDS:** High-throughput sequencing, near-natural forest management, mixed
forest, rhizosphere bacteria, soil properties, tree species

**INTRODUCTION**

As a gateway for plants to absorb water and nutrients, and to interact with the soil
matrix, the rhizosphere zone plays a pivotal role in plant life and soil ecosystems (1).
In nature, the interactions between root and soil are very complex, in which
rhizosphere microorganisms actively participate (2). The properties of rhizosphere soil
are largely determined by the interaction of soil, plants, and ~~root-related~~
microorganisms. Some rhizosphere microorganisms can fix nitrogen, ~~release~~

[revised manuscript text omitted]

the website (<https://huttenhower.sph.harvard.edu/galaxy/>).

**RESULTS**

**Chinese fir in different mixed forests**

*Physicochemical properties of rhizosphere soil*

The rhizosphere soil properties of Chinese fir displayed pronounced changes
under near-natural forest management (Table 1). The TP content was significantly
higher, reaching $0.47 \text{ g}\cdot\text{kg}^{-1}$, from rhizosphere bacterial communities of SHX_S than
either S_S or SD_S, but vice versa for the AP content, being only $6.5 \text{ mg}\cdot\text{kg}^{-1}$ in
SHX_S. By contrast, the other five indicators (pH, SOM, SOC, TN, AN) showed no
significant differences among the three stand types ($P > 0.05$). Compared with S_S,

the rhizosphere soil pH of SD_S was slightly higher at 4.7 ± 0.1 , whereas that of
 SHX_S was lower, at just 4.4 ± 0.1 . These results revealed that the introduction of
 *Michelia hedyosperma* and *Castanopsis hystrix* reduced the content of AP in the
 Chinese fir rhizosphere soil.

**Table 1 Rhizosphere soil properties of Chinese fir trees in different mixed forests**

Mixed patterns	pH	SOM(g·kg ⁻¹)	SOC(g·kg ⁻¹)	TN(g·kg ⁻¹)	AN(mg·kg ⁻¹)	TP(g·kg ⁻¹)	AP(mg·kg ⁻¹)
S_S	4.5 ± 0.14a	35.24 ± 1.56a	20.44 ± 0.91a	1.57 ± 0.05a	181.02 ± 11.67a	0.33 ± 0.01b	10.94 ± 1.58a
SHX_S	4.36 ± 0.17a	45.67 ± 11.39a	26.50 ± 6.61a	1.87 ± 0.56a	223.31 ± 70.78a	0.47 ± 0.08a	6.50 ± 1.56b
SD_S	4.66 ± 0.14a	40.53 ± 6.85a	23.51 ± 3.97a	1.76 ± 0.36a	178.00 ± 28.57a	0.32 ± 0.03b	10.85 ± 2.79a

233 **Notes:** Values are the means ± SD (Standard Deviation) (n=3). Different letters indicate significant
 differences ($P < 0.05$). Abbreviations: SOM, soil organic matter; TN, total nitrogen; TP, total
 phosphorus; AN, available nitrogen; AP, available phosphorus; S_S, Chinese fir in pure
 *Cunninghamia lanceolata* forest; SHX_S, Chinese fir in *Cunninghamia lanceolata*–*Castanopsis*
 *hystrix*–*Michelia hedyosperma* mixed forest; SD_S, Chinese fir in *Cunninghamia lanceolata*–
 *Castanopsis fissa* mixed forest.

*Composition and diversity of bacterial communities*

There were 4765, 5322, and 5163 OTUs in the rhizosphere soil of S_S, SHX_S,
 and SD_S, respectively, with 3148 OTUs shared by the three stand types (Fig. 1c).
 The numbers of unique OTUs in the rhizosphere soil of S_S, SHX_S, and SD_S were
 647, 1007, and 1008, respectively. More unique OTUs occurred in the rhizosphere
 soil of SHX_S and SD_S. We then assessed the taxonomic distributions of these
 bacterial OTUs (Fig. 1a and b). Whether at the phylum or the genus level, no

significant differences in the composition of bacterial communities were detected
among the rhizosphere soil of Chinese fir trees growing in different mixed forests.
However, the rhizosphere soil bacterial communities of Chinese fir in SHX and SD
contained a higher total absolute abundance than did pure forest (S), reaching 564063
and 618759 respectively. At the phylum level, seven phyla with an absolute
abundance proportion greater than 1% were identified: *Acidobacteria*, *Proteobacteria*,
*Actinobacteria*, *Chloroflexi*, *Planctomycetes*, *Gemmatimonadetes*, *Bacteroidetes*, and
No_rank (Fig. 1a). Both *Acidobacteria* and *Proteobacteria* dominated the bacterial
communities in the rhizosphere, accounting for 71.6%–76.5% of total absolute
abundance. At the genus level (Fig.1b), 14 genera were detected with a proportion
higher than 1%, of which No_Rank, Gp2, Unassigned, and Gp1 were the dominant
genera. The cumulative absolute abundance of the four genera accounted for 72.9%–
74.2% of total absolute abundance.

One-way ANOVA was conducted at the phylum and genus level on the
rhizosphere bacterial communities to find bacteria with significant differences. The
results showed the absolute abundance of *Planctomycetes*, *Actinobacteria*,
*-Clostridium*, *Novosphingobium*, and *Pseudonocardia* were significantly higher in
SD_S compared to S_S. *Verrucomicrobia* and *Actinobacteria* were significantly
higher in SHX_S than S_S, while the absolute abundance of *Staphylococcus* and
*Rhodococcus* was significantly higher in S_S than SHX_S and SD_S, respectively
(Fig. S1).

To investigate the differences among species in the Chinese fir rhizosphere soil

bacterial communities of different mixed patterns, we used the linear discriminant
 analysis effect size (Lefse) algorithm (LDA log score >3.0 and P < 0.05, the length of
 the bar chart represents the impact of different species) (Fig. 1d). *Staphylococcaceae*
 got the highest LDA score (3.779) in rhizosphere bacteria in S_S, indicating that it
 was considerably enriched in S_S. *Actinobacteria* (4.086) had the highest LDA scores
 and were considerably enriched in SHX_S, whereas *Planctomycetaceae* (4.803) was
 significantly enriched in SD S.

**Fig. 1** Rhizosphere bacterial community composition and bacteria with significant
differences in Chinese fir at different mixed patterns. **a** Rhizosphere bacterial
community composition on phylum level. **b** Rhizosphere bacterial community
composition on genus level. **c** Bacteria OUTs distribution in the Chinese fir
rhizosphere soil. **d** Taxa enriched in Chinese fir at different mixed patterns are
indicated with LDA scores, respectively. Only taxa with LDA values greater than 2.2
($P < 0.05$) are shown.

Except for the Simpson index, the three other diversity indexes of the rhizosphere
bacterial community were highest in SHX_S (Fig. 2a). Changes in the diversity index
from pure forest to mixed forest revealed that near-natural forest management had
improved the species diversity and uniformity of the rhizosphere bacterial community
of Chinese fir. The PCA axis 1 and 2 respectively explained described 45.59% and
15.19% of the distribution of Chinese fir rhizosphere bacteria (Fig. 2b). In this PCA
plot, the SD_S samples separated from those of S_S and SHX_S demonstrated a great
difference between SD_S and the other two stand types.

~~We analyzed the interaction network of Chinese fir rhizosphere bacteria in~~
~~different mixed patterns. Pearson correlations between genera were calculated based~~
~~on their absolute abundance in soil samples of Chinese fir in different mixed forests.~~
In the analysis of the top 40 bacterial species at the genus level, we found that there
were 59 edges and 33 nodes in S_S, 43 edges and 35 nodes in SD_S, 36 nodes and 47
edges in SHX_S (Fig. 2c and Table S1). These results indicated that the interaction
network in SD_S and SHX_S was relatively simple compared to the pure forest, but
the complexity between bacteria was elevated. The key microbial groups belonged

mainly to *Acidobacteria*, *Proteobacteria*, *Planctomycetes*, and *Actinobacteria*, and the
 centrality of *Acidobacteria* became lower and that of *Proteobacteria* became higher in
 SD_S and SHX_S compared to S_S.

**Fig. 2** Bacterial diversity and bacterial interaction networks in Chinese fir at different
mixed patterns. **a** Alpha diversity of rhizosphere bacterial communities. **b** Principal
component analysis (PCA) of bacterial communities in the rhizosphere of Chinese fir.
**c** The interaction network of dominant microbiota at the genus level (top 40) in
rhizosphere soil of S_S, SD_S, and SHX_S. And the different colors indicate the
corresponding taxonomic assignment at the phylum level. The edge color represents
positive (red) and negative (blue) correlations. Only significant interactions are shown
($r > 0.6$; $P < 0.05$).

*Relationships between soil physicochemical and bacteria community structure*

The SOM, TN, and SOC had significant positive associations with *Bacteroidetes*
($P < 0.05$) according to the heatmap of tested correlations between the absolute
abundance of rhizosphere bacterial communities and soil physicochemical parameters
(Fig. 3a). The AN was significantly positively correlated with *Gemmatimonadetes* yet
negatively correlated with *Chloroflexi* ($P < 0.05$). Although TP had a significant
positive correlation with *Verrucomicrobia*, AP had a significant negative correlation
with it ($P < 0.05$). Redundant analysis (RDA) at the phylum level (Fig. 3b)
demonstrated that SOM, TN, AN, TP, and pH values were the main rhizosphere soil
property factors that significantly impacted the soil rhizosphere microbes. RDA axis 1
and 2 respectively captured 29.97% and 20.33% of the variability in rhizosphere soil
bacteria. At the phylum level, SOM exerted the most substantial effect upon the
bacterial community in the rhizosphere of Chinese fir.

**Fig. 3** Correlation relationships (A) and redundancy analysis (RDA) were carried out
 (B) between Chinese fir rhizosphere bacteria and rhizosphere soil physicochemical
 properties in different plantation stand types, at the phylum level. * : $P < 0.05$.

Different tree species in the same mixed forest

*Physicochemical properties of rhizosphere soil*

As Table 2 shows, the AP content of rhizosphere soil in SHX_S was significantly
 higher (by 2.93 and 2.84 $\text{mg}\cdot\text{kg}^{-1}$ respectively) than that in SHX_H and SHX_X. The
 pH in the rhizosphere soil of SHX_X was the highest, reaching 4.60, and that of
 SHX_S the lowest, at only 4.36. For the content of SOM, SOC TN, TP, and AN, each
 exhibited a trend of SHX_S > SHX_H > SHX_X, whereas the trend for pH shifted to
 SHX_S < SHX_H < SHX_X. Except for pH and TP, the SOM, SOC, TN, AN, and AP
 content values in the rhizosphere soil were all lower in SD_S than SD_D. The pH
 value of SD_D was 4.47, which this lower than that of SD_S (4.66).

**Table 2** Rhizosphere soil physicochemical properties of different tree species in SHX

and SD

Tree species	pH	SOM(g·kg ⁻¹)	SOC(g·kg ⁻¹)	TN(g·kg ⁻¹)	AN(mg·kg ⁻¹)	TP(g·kg ⁻¹)	AP(mg·kg ⁻¹)
SHX_S	4.36 ± 0.17a	45.67 ± 11.39a	26.50 ± 6.61a	1.87 ± 0.56a	223.31 ± 70.78a	0.47 ± 0.08a	6.50 ± 1.56a
SHX_H	4.40 ± 0.12a	36.27 ± 7.59a	21.04 ± 4.4 a	1.58 ± 0.21a	192.67 ± 25.38a	0.46 ± 0.07a	3.56 ± 0.29b
SHX_X	4.60 ± 0.14a	33.81 ± 2.55a	19.61 ± 1.48a	1.26 ± 0.28a	186.63 ± 24.37a	0.43 ± 0.07a	3.66 ± 0.65b
SD_S	4.66 ± 0.14a	40.53 ± 6.85a	23.51 ± 3.97a	1.76 ± 0.36a	178.00 ± 28.57a	0.32 ± 0.03a	10.85 ± 2.79a
SD_D	4.47 ± 0.09a	42.93 ± 3.76a	24.90 ± 2.18a	1.90 ± 0.05a	203.03 ± 23.02a	0.32 ± 0.02a	13.69 ± 0.87a

349 **Notes:** Values are the means ± SD (Standard Deviation) (n=3). SHX_S: Chinese fir in SHX;

SHX_H: *Castanopsis hystrix* in SHX; SHX_X: *Michelia hedyosperma* in SHX; SD_S: Chinese fir

in SD; SD_D: *Castanopsis fissa* in SD. Abbreviations: Different letters indicate significant

differences ($P < 0.05$).

*Composition and diversity of bacterial communities*

The bacterial communities of SHX_S, SHX_H, and SHX_X contained 5322, 5619,

and 5263 OTUs, respectively (Fig. 4c). Overall, 3603 OTUs were shared by the three

tree species, whereas 800, 860, and 569 OTUs were respectively unique to the

rhizospheres of SHX_S, SHX_X, and SHX_H. The total absolute abundance of the

bacterial communities of SHX_S was 181.74 % and 148.31% higher than that of

SHX_X and SHX_H, respectively (Fig. 4a). At the phylum level, seven phyla were

detected, namely *Acidobacteria*, *Proteobacteria*, *Chloroflexi*, *Actinomycetes*,

*Planctomycetes*, *unassigned*, *Gemmatimnadetes*, No-rank, and *Bacteroides*. At the

genus level, a total of 13 genera with a proportion greater than 1% in the sample were

detected (Fig. 4b). Among them, the dominant ones were Unassigned, Gp2, Gp1, and

No_Rank, and the cumulative absolute abundance of these four genera constituted
72.0%–72.9% of the total absolute abundance.

There were 5163 and 5308 bacterial OTUs in the rhizosphere soil samples of
SD_D and SD_S, respectively (Fig. 5c). The bacterial communities of SD_S and
SD_D had unique 1315 and 1170 OTUs, respectively. As evinced by Fig. 5a and b,
the composition of rhizosphere bacterial communities of SD_S and SD_D were
similar. As for SHX, *Acidobacteria* and *Proteobacteria* were the dominant phyla in
SD, with the former having the highest absolute abundance value. The cumulative
absolute abundance of *Acidobacteria* and *Proteobacteria* was 472 340 and 675 679,
this accounting for 76.34% and 72.95% of the total absolute abundance, respectively.
The absolute abundance of SD_D was significantly higher than that of SD_S, but the
absolute abundance of SD_S was only 0.67 times that of SD_D (Fig. 5a). At the genus
level, 10 genera with a proportion greater than 1% were detected: No rank, Gp2,
unassigned, Gp1, *Rhodoplanes*, *Rhizomicrobium*, *Bradyrhizobium*, *Burkholderia*, Gp6,
and Gp3 (Fig. 5b). Of these, four genera belonged to *Acidobacteria*, namely Gp2, Gp1,
Gp6, and Gp3. Absolute abundance for the same genus was ranked as SD_S < SD_D.
The results of one-way ANOVA (Fig. S2) showed that among the bacterial phyla and
genera with significant differences in SHX ($p < 0.05$), the absolute abundance of
*Actinobacteria*, *gp1*, *Kitasatospora*, *Paenibacillus*, *Rhodoplanes*, *Streptomyces* of
SHX_S rhizobacteria was significantly higher than that of SHX_H and SHX_X. The
absolute abundance of *Thaumarchaeota*, *Gemmata*, *Gp7*, *longilinea*, *longispora* in the
rhizosphere soil of SHX_X was the highest. As for the SD, it can be seen that the

phyla and genera with significant differences, such as *Proteobacteria*, *Planctomycetes*,
*Dactylosporangium*, *Gaiella*, Gp1, *Granulicella*, Gp3, *Novosphingobium*, *Terriglobus*,
*Rickettsia*, *Roseiarcus* and *Streptacidiphilus* all showed a greater absolute abundance
of SD-D than SD-S (Fig. S3).

We calculated the LDA values among different tree species in SHX and SD. The
results showed that in the SHX, *Anaerolineaceae* (*Chloroflexi*) enriched in SHX_X
with the highest LDA score (3.571); *Actinobacteria* was enriched in SHX_S with an
LDA value of 4.124. The enriched genera in SHX_H was *Salmonella*, up to 3.202
(Fig. 4d). In SD, *Proteobacteria* and *Actinobacteria* were predominantly enriched in
SD_D with more differentially significant species, including *Roseiarcus*, *Rhizobiales*,
*Sphingomonaceae*, *Oxalobaceae*, *Comamonadaceae*, *Hyphomicrobiaceae*,
*Burkholderials* (*Proteobacteria*), *Gaiella*, *Jatrophihabitans*, and *Streptacidiphilus*
(*Actinobacteri*). *Telmatospirillum* (*Proteobacteria*) was enriched in SD_S (Fig.5d).

**Fig. 4** Rhizosphere bacterial community composition and bacteria with significant
 differences in different tree species at SHX. **a** Rhizosphere bacterial community
 composition on phylum level. **b** Rhizosphere bacterial community composition on
 genus level. **c** Bacterial OUTs distribution in rhizosphere soil. **d** Taxa enriched in
 different tree species at SHX are indicated with LDA scores, respectively. Only taxa
 with LDA values greater than 2.2 ($P < 0.05$) are shown.

**Fig. 5** Rhizosphere bacterial community composition and bacteria with significant
 differences in different tree species at SD. **a** Rhizosphere bacterial community
 composition on phylum level. **b** Rhizosphere bacterial community composition on
 genus level. **c** Bacteria OUTs distribution in rhizosphere soil. **d** Taxa enriched in
 different tree species at SD are indicated with LDA scores, respectively. Only taxa
 with LDA values greater than 2.2 ($P < 0.05$) are shown.

Species richness of rhizosphere bacteria in SHX_H exceeded that of SHX_S,
whereas that of SHX_X was the lowest (Fig.6a). The Shannon index of SHX_X was
the greatest, indicating its rhizosphere bacterial community had the highest diversity.
The PCA's first axis explained 49.17% of the distribution of rhizosphere bacteria, and
its second axis explained 22.19% of it (Fig. 6b). The PCA demonstrated that
rhizosphere bacteria of SHX_H and SHX_X were more similar in rhizosphere
bacterial community composition. Excluding the Shannon index, the Chao1, ACE,
and Simpson index the rhizosphere bacteria of SD_S were all higher than those of
SD_D, indicating that the former's diversity of rhizosphere bacteria was lower than
the latter's (Fig. 7a). PCA confirmed that samples from these two groups showed a
clear separation (Fig. 7b). In the network analysis of the absolute abundance of
bacteria in the top 40 at the genus level in SHX, there were 36 nodes and 47 edges in
SHX_S, 33 nodes and 38 edges in SHX_H, and 32 nodes and 40 edges in SHX_X
(Fig.6c and Table S2). Compared to SHX_S, the complexity of the interaction
network in SHX_X and SHX_H, and the average degree decreased. The centrality of
*Proteobacteria*, and *Chloroflexi* became higher in rhizobacteria of SHX_H compared
to SHX_S, while the centrality of *Planctomycetes* and *Actinobacteria* were higher in
rhizobacteria of SHX_X. SD_S and SD_D had the same number of nodes (35), and the edges
of SD_D were higher than those of SD_S, indicating that the more complicated interaction
network of SD_D (Fig.7c and Table S3)

**Fig. 6** Bacterial diversity and bacterial interaction networks in different tree species at
 SHX. **a** Alpha diversity of rhizosphere bacterial communities. **b** Principal component
 analysis (PCA) of bacterial communities in the rhizosphere soil. **c** The interaction
 network of dominant microbiota at the genus level (top 40) in rhizosphere soil of
 SHX_S, SHX_H, and SHX_X. And the different colors indicate the corresponding
 taxonomic assignment at the phylum level. The edge color represents positive (red)
 and negative (blue) correlations. Only significant interactions are shown ($r > 0.6$; $P <$
 0.05).

**Fig. 7** Bacterial diversity and bacterial interaction networks in different tree species at
 SD. **a** Alpha diversity of rhizosphere bacterial communities. **b** Principal component
 analysis (PCA) of bacterial communities in the rhizosphere soil. **c** The interaction
 network of dominant microbiota at the genus level (top 40) in rhizosphere soil of
 SD_S and SD_D. And the different colors indicate the corresponding taxonomic
 assignment at the phylum level. The edge color represents positive (red) and negative
 (blue) correlations. Only significant interactions are shown ($r > 0.6$; $P < 0.05$).

[revised manuscript text omitted]

The absolute abundance of *Planctomyces*, *Actinobacteria* was significantly
higher in rhizosphere soil of SD_S than in S_S, while the absolute abundance of
*Verrucomicrobia*, *Actinobacteria* was significantly higher in rhizosphere soil SHX_S
than in S_S (Fig. S2 and S3). Lefse results corroborated ANOVA results (Fig. 1d),
*Actinobacteria* were significantly enriched in SHX_S, while *Planctomycetaceae* were
significantly enriched in SD_S. This suggested that they may play an important role in
near-natural cultivation of Chinese fir and the bacteria that play different roles in
different mixed patterns. Genera under the *Planctomycetaceae* had the potential for
production of exopolysaccharides and lipopolysaccharide, which was beneficial to the
formation of microaggregates in soil (69). *Actinobacteria* mainly promote soil decay of

animal and plant remains to decay. Some reported *Actinobacteria* can produce related
enzymes for decomposing lignin and cellulose (70). For example, *Thermobifida* was
found to bind efficiently extracellular enzymes with xylanase(71). *Opitutus* belonging
to *Verrucomicrobia* was known as a specialized anaerobic bacterium and was
previously associated with nitrogen fixation in rice fields under strictly anaerobic
conditions (72)

Abundance levels determine the functional role of bacteria in complex
communities, and microbial communities featuring greater diversity are thought to be
more resistant to pathogen invasion and more stable (73, 74). Our results showed that
the introduction of *Michelia hedyosperma* and *Catanoposis hystrix* influenced the
bacterial community in the rhizosphere of Chinese fir. This is similar to the findings
of two other studies (75, 76), where the introduction of broad-leaved tree species into
pure plantations changed the composition and structure of their soil microbial
community. Studies have reported that the introduction of broad-leaved tree species in
pure Chinese fir plantations improves soil quality by increasing organic matter,
effective nutrients, and soil microbial activity (48, 77). This may be because, after
introducing them, the average amount of annual litter increased considerably, and its
composition changed, which hastened the input of nutrients from the ground surface
(78, 79), and indirectly led to changed content of nutrients in the soil.

Microbial network analysis, by identifying the species with high connectivity
throughout the network or the position located within the species module, allows to
obtain the key species and the more important species in the entire network, and these

species may have a determining role in the structure and function of the microbial
community (80). The topological characteristics of the co-occurrence network showed
that the community complexity of rhizosphere bacteria of SD_S and SHX_S was
increased (Table S1 and Fig. 2c). The key node nodes belonged mainly to
*Acidobacteria*, *Proteobacteria*, *Planctomycetes*, and *Actinobacteria*, which may
perform key ecological functions associated with near-natural cultivation.

The diversity of tree species indirectly affects soil microbial diversity (81), and
the higher bacterial diversity in the rhizosphere of SHX_S may have arisen from the
higher diversity of its aboveground tree species and more complex composition of
litter, which indirectly impacts rhizosphere bacterial diversity. Furthermore, according
to the type and quantity of vegetation litter, in addition to root exudates, especially
their chemical characteristics, they can selectively stimulate the growth of soil
microorganisms, thereby influencing the characteristics of the microbial community
(82, 83).

**Effects of tree species on the structure and diversity of bacteria in rhizosphere**
**soils**

In the same mixed stand type, the rhizosphere bacterial community of
broad-leaved tree species and Chinese fir had a similar composition at the phylum and
genus level, similar to other reports (37, 84, 85), although there were stark differences
in their absolute abundance. In SHX, the absolute abundance of rhizosphere bacteria
declined successively across Chinese fir, *Michelia hedyosperma*, and *Catanopsis*

*hystrix*, and this difference was mainly reflected in *Actinobacteria* (Fig S2 and Fig.
4d). The results of network analysis showed that the diversity and complexity of
rhizosphere bacterial communities of Chinese Fir were higher than those of
*Castanopsis hystrix* and *Michelia hedyosperma* (Fig.6b). The rhizosphere bacteria
richness was higher under *Catanoposis hystrix* than Chinese fir, being lowest under
*Michelia hedyosperma*. Conversely, the Shannon index for rhizosphere bacteria of
*Michelia hedyosperma* was the highest, revealing that this species supported the
highest level of bacterial community diversity. In SD, the absolute abundance of
bacteria was lower under Chinese fir than *Catanoposis fissa*, but this trend was
reversed for their Shannon index. *Proteobacteria* were enriched in rhizobacteria of
SD_D and its rhizosphere bacterial community complexity was higher than that of
SD_S (Fig S3 and Fig. 5d). According to Fig.5d, the absolute abundance of bacteria
related to nitrogen fixation, such as *Burkholderiales* and *Rhizobiales*, in SD_S was
higher than that in SD_S (86, 87). *Sphingomonaceae* was a member of hydrocarbon

[revised manuscript text omitted]

*Proceedings of the National Academy of Sciences of the United States of America*
103:626.<https://doi.org/10.1073/pnas.0507535103>.
- 109. Zhang DH, Lin KM, Li BF. 2011. Phosphorus characteristics in rhizosphere soil of
*Cunninghamia lanceolata*, *Pinus massoniana* and their mixed plantation. *Chinese Journal of
Applied Ecology*
doi:<https://doi.org/CNKI:SUN:YYSB.0.2011-11-006:2815-2821>.<https://doi.org/CNKI:SUN:YYSB.0.2011-11-006>.
- 110. Fan J. 2015. Stand structure influence on phosphorus functional fractions and vegetative
organ nitrogen and phosphorus stoichiometric ratio in *cunninghamia lanceolata* plantations.
Master. Jiangxi Agricultural University.
- 111. Wen Y, Li H, Zhou X, Zhu H, Li Y, Cai D, Jia H, Huang X, You Y. 2019. Effects of uneven-aged
*Pinus massoniana* × *Castanopsis hystrix* mixed plantations on structural and functions of soil
microbial community. *Guangxi Science*
26:188-198.<https://doi.org/10.13656/j.cnki.gxkx.20190419.001>.
- 112. Sattar M, Gaur A. 1987. Production of auxins and gibberellins by phosphate-dissolving
microorganisms. *Zentralblatt für Mikrobiologie*
142:393-395.[https://doi.org/10.1016/S0232-4393\(87\)80086-0](https://doi.org/10.1016/S0232-4393(87)80086-0).

- 113. Khan MS, Zaidi A, Wani PA. 2007. Role of phosphate-solubilizing microorganisms in
sustainable agriculture—a review. *Agronomy for sustainable development*
27:29-43.<https://doi.org/10.1051/agro:2006011>.
- 114. Kalayu G. 2019. Phosphate solubilizing microorganisms: promising approach as biofertilizers.
*International Journal of Agronomy* 2019:7.<https://doi.org/10.1155/2019/4917256>.

Dear reviewers,

Thank you very much for your comments and professional advice. These opinions help to improve the academic rigor of our article. Based on your suggestion and request, we have made corrected modifications to the **Marked-up manuscript (without Figures)**. Each figure has been uploaded as a separate file. We hope that our work can be improved again. The responses to the reviewer's comments are marked in red and presented following:

Answers to Reviewer #2:

1. I strongly recommend a graphical abstract or diagram to clearly indicate the symbols/abbreviations used for the forest types and trees. It is very difficult to keep track of this, especially as the abbreviations are not acronyms.

The author's answer: Thank you for your suggestion. As suggested by the reviewer, this has been clarified in the revised version of the manuscript, and **Figure 9** has been included in the Materials and Methods section as a new Figure to clearly indicate the symbols/abbreviations used for the forest types and trees. The Figure 9 was bellow:

Fig. 9 Study area (a) and sampling site (b) in Pingxiang, Guangxi Province of China. Abbreviations were showed in (c).

2. In Lines 249-251, it is not clear what absolute abundance here means. If it means a total number of sequences obtained per sample per forest, then that is a technical issue rather than a biological one. It is typical to conduct equimolar pooling of multiplexed samples prior to loading the library onto the MiSeq instrument. If that was done, it here did not work. It is one thing to talk about the unique approach to the "absolute abundance" of individual bacterial taxa (when the total DNA from each library is equal), yet another entirely to speak of the "absolute abundance" of soil samples in general. A key potential technical explanation for observed differences is that different forest soils have different amounts of PCR inhibitors. There can also be processing issues if the samples were not processed in a randomized fashion. [This same issue arises in the tree-specific analyses later in the manuscript. Just a couple of sentences.]

The author's answer: Thanks for your question, our explanation is as follows: The **absolute quantification** of 16S rRNA was performed externally, by the company Genesky Biotechnologies Inc. (Shanghai, 201315, China). First, we constructed and sequenced 16S amplicon libraries by adding a certain amount of **spike-in standards** artificially synthesized sequences to the sample DNA, and then drew **a standard curve** based on the number of spike-in standards 16S amplicon reads and their absolute copies. Finally, the absolute copies of the 16S rRNA gene of the species in the range of the standard curve in the sample were calculated. **Therefore, the absolute abundance is actually the absolute copies.** The total absolute abundance value of each sample obtained by this method is different, which is different from the previous relative abundance determination method. Relevant references are as follows:

1. Bolyen E, et al. Reproducible, interactive, scalable and extensible microbiome data science using QIIME 2. *Nature Biotechnology*. 2019,37(8):852-857.
2. Callahan BJ, et al. DADA2:high-resolution sample inference from Illumina amplicon data. *Nature methods*, 2016,13(7):581-3.
3. Jiang S Q , et al. High-throughput absolute quantification sequencing reveals the effect of different fertilizer applications on bacterial community in a tomato cultivated coastal saline soil[J]. *Science of The Total Environment*, 2019, 687:601-609.

3. Methods: In *Microbiology Spectrum*, the Methods come after the Discussion section.

The author's answer: We thank the reviewer for pointing this out. We have revised the position of the Methods to put it after the Discussion section. **Please see Pages 29-33 of the Marked-up manuscript.**

4. Results- Line 405: There are several places in the manuscript wherein "OTU" is misspelled as "OUT"

The author's answer: We've changed "OUT" to "OTU" on Figure 4 and 5

5. In general, I recommend not reporting in the text the:

- 1) percent variance explained by PCA axes; this information is in the figures
- 2) specific numbers of nodes and edges in networks for specific forests and trees

3) etc.

The results are summarized in the text and the specifics are in the figures. If a reader desires to focus on them, they can consult the figures. At present, the Results section is a very difficult read. More narrative reporting and less specific recounting of results that are also presented in figures and tables would make the manuscript easier to read and digest.

The author's answer: We thank the reviewer for pointing this out. We have made some changes in the text, including reducing the specific description of data and adding narrative reporting. Some modifications are as follows:

(1) We changed the sentence "There were 4765, 5322, and 5163 OTUs in the rhizosphere soil of S_S, SHX_S, and SD_S, respectively, with 3148 OTUs shared by the three stand types" into "The rhizobacterial OTUs of Chinese fir under mixed treatment increased by 1.08-1.12 times compared with pure forest, among which SHX_S was the highest" (Page 12, line 250-252);

(2) We replaced "The PCA axis 1 and 2 respectively explained described 45.59% and 15.19% of the distribution of Chinese fir rhizosphere bacteria (Fig. 2b)." with "Principal components analysis (PCA) showed that 60.78% of the variance was explained by the first two components (Fig. 2b)" (Page 14, line 299-303).

For more modifications, please refer to the Marked-up manuscript.

6. Throughout the manuscript, is Simpson being reported as the Simpson Index, or is it being reported as 1- Simpson or Inverse Simpson? Please be explicit about this in the Methods.

The author's answer: This suggestion is appreciated. We have specified that the Simpson was being reported as the Simpson Index in the Methods section (Page 32, lines 698-700).

Considering the Reviewer's suggestion, We have made the appropriate modifications. Special thanks to you for your good comments.

Answers to Reviewer #3:

1. Considering rephrasing the topic for clarity "Effect of Near-natural Forest Management and Tree Species on Chinese Fir Plantation Rhizosphere and Bacterial Communities"

The author's answer: Thank you for the suggestion. After careful consideration, we have changed "Near-Natural Forest Management and Tree Species Affect the Chinese fir plantation Rhizosphere Soil Properties and Rhizosphere Bacterial Communities" to "Rhizosphere Bacterial Communities Response to Near-Natural Forest Management and Tree Species in Chinese fir plantation" (Page 1, lines 3-4)

2. L15-The font size for the word "Abstract" is not consistency

The author's answer: Thank you for underlining this deficiency. We have changed the font size for the word "Abstract" (Page 1, line 17)

3. L26- remove the word "did"

The author's answer: We have removed the word "did" (Page 1, line 28).

4. L26 and L27- replace the word "enriched" with "abundant"

The author's answer: We have replaced the word "enriched" with "abundant"(Page 2, line 28).

5. L30- replace nitrogen fixation-related with "nitrogen fixing"

The author's answer: We have replaced "nitrogen fixation-related bacteria" with "Bacteria related to nitrogen-fixing"(Page 2, line 33).

6. L31-replace "present" with "abundant"

The author's answer: We have replaced "present" with "abundant"(Page 2, line 34)

7. L35-L38- rephrase for clarity"

The author's answer: We have replaced " Our results showed that near-natural management of introduced broad-leaved tree species for 12 years can drive the alteration of physicochemical properties and bacterial community structure and composition of rhizosphere soil in Chinese fir plantations, with tree species identity further shaping the rhizosphere soil bacterial community." with " Our results demonstrated that, in Chinese fir plantations, 12 years of near-natural management of introduced broad-leaved tree species can drive the alteration of the physicochemical characteristics, bacterial community structure, and composition of rhizosphere soil, with tree species identity further influencing the rhizosphere soil bacterial community." (Page 2, lines 37-42)

8. L60- replace "root-related" with "rhizospheric"

The author's answer: We have replaced "root-related" with "rhizospheric" (Page 4, line 68).

9. L61-62- replace "release plant-avaliable phosphorus and" with "solubilize"

The author's answer: We have replaced "release plant-available phosphorus " with "solubilize"(Page 4, line 69).

10. L69- replace "reduces" with "has also been found to reduce"

The author's answer: We have replaced "reduces" with "has also been found to reduce"(Page 4, line 78).

11. L80- Remove "secretion", it has the same meaning as exudation

The author's answer: We have removed the word "secretion" (Page 5, line 89).

12. L95- Remove the fullstop

The author's answer: We have removed the full stop (Page 5, line 104).

13. Li97-98 what makes your work different from these reported research

The author's answer: Our study differs in that it focuses on the changes in soil nutrients and soil bacterial communities at the **rhizosphere** of Chinese fir, whereas previous studies have mostly focused on non-rhizosphere soil nutrients and microbial communities.

14. L174- Remove the double bracket

The author's answer: We have removed the double bracket (Page 31 , line 667).

15. L197-198- Font size here is smaller than others

The author's answer: We have amended the font size here to be the same as in the rest of the text (Page 32, lines 689-691).

16. L223-224- why is the _ here, it was not in the code written in the materials and methods

The author's answer: We apologize for our oversight in not including the meaning of this code in the Materials and Methods. We have added **a new Figure (Figure 9b and c)** to clarify the meaning of the _. The Figure 9 was bellow:

Fig. 9 Study area (a) and sampling site (b) in Pingxiang, Guangxi Province of China.

Abbreviations were showed in (c).

17. L224- check the _ here as well as line 225 for clarity

The author's answer: We have added a new Figure (Figure 9) to clarify the meaning of the _.

18. L250- Replace with "the"

The author's answer: We have replaced "did" with "the" (Page 12, line 261).

19. L254- Define the _ properly for clarity

The author's answer: Unlike questions 16 and 17, No_rank here denotes bacteria that do not have explicit classification information or classification names at some classification level.

20. L264- Since all these have been abbreviated in the materials and methods sections, the same abbreviation should be used in the result section for consistency and clarity, kindly adjust this throughout the manuscript. To make it easier, since you have done a lot of analysis, you can change the coding in the materials and methods section to suit that in the result and figures

The author's answer: We apologize that some of the references to abbreviations were not included in the Material and Methods, but only in the notes below the charts. We have added a new Figure (Figure 9) to clarify the meaning of all the abbreviations.

Fig. 9 Study area (a) and sampling site (b) in Pingxiang, Guangxi Province of China. Abbreviations were showed in (c).

21. L274- Remove excess space

The author's answer: We have removed excess space (Page 14, line 290).

22.L286- Remove extra space before the paragraph

The author's answer: We have set the paragraph formatting to indent the first line by two characters (Page 14, line 306).

23. L294-296 This is rather materials and methods not results

The author's answer: Thank you for your suggestion. As suggested by the reviewer, we have deleted this sentence, added "network" on Page 15, Line 309, and added "(using Pearson correlation)" on Page 33, Line 708.

24.L301 Add observed in the study

The author's answer: Thank you for your comments, we have added "observed in the study" (Page 15, line 315)

25.L309-315 Add key before the A, do the same for similar figures

The author's answer: We deeply appreciate the reviewer's suggestion. We have added brackets to the annotations in all similar diagrams where the combination diagrams were labeled a.b.c, etc. For example, modifying "a" to "(a)" (Figure 2).

We appreciate for Reviewers' warm work earnestly and hope that the correction will meet with approval. Once again, thank you very much for your comments and suggestions. Look forward to hearing from you!

Yours sincerely,

Aiguo Duan

21st Oct, 2022

Chinese Academy of Forestry

November 30, 2022

Prof. Aiguo Duan
Chinese Academy of Forestry
Beijing
China

Re: Spectrum02328-22R1 (Near-Natural Forest Management and Tree Species Affect the Chinese fir plantation Rhizosphere Soil Properties and Rhizosphere Bacterial Communities)

Dear Prof. Aiguo Duan:

Thank you for submitting your manuscript to Microbiology Spectrum. As you will see your paper is very close to acceptance. Before acceptance, see the remaining comments below by Reviewer #2, especially comment #2 regarding interpretation of the Simpson index.

As these revisions are quite minor, I expect that you should be able to turn in the revised paper in less than 30 days, if not sooner.

When submitting the revised version of your paper, please provide (1) point-by-point responses to the issues raised by the reviewers as file type "Response to Reviewers," not in your cover letter, and (2) a PDF file that indicates the changes from the original submission (by highlighting or underlining the changes) as file type "Marked Up Manuscript - For Review Only". Please use this link to submit your revised manuscript. Detailed instructions on submitting your revised paper are below.

Link Not Available

Sincerely,

Kevin R. Theis

Reviewer comments:

Reviewer #2 (Comments for the Author):

I appreciate the authors' efforts in revising the manuscript. A few minor points remain.

1) The new title may be more clear as:

Response of Rhizosphere Bacterial Communities to Near Natural Forest Management and Tree Species Within Chinese Fir Plantations

2) The authors indicate that Simpson index (not inverse Simpson or $1 - \text{Simpson}$) is used, which means that the lower the value, the higher the alpha diversity. The data from this alpha diversity index are then in accordance with the other diversity indices used (Figure 2A). This should be corrected before publication.

3) In the Legend for Figure 9C, correct the spelling for "abbreviations".

Reviewer #3 (Comments for the Author):

The paper has been properly revised to my satisfaction

Preparing Revision Guidelines

Please return the manuscript within 60 days; if you cannot complete the modification within this time period, please contact me. If you do not wish to modify the manuscript and prefer to submit it to another journal, please notify me of your decision immediately so that the manuscript may be formally withdrawn from consideration by Microbiology Spectrum.

Dear reviewers,

Thank you for your decision and constructive comments on my manuscript. We have carefully considered the suggestion of the Reviewer and made some changes. We have made corrected modifications to **the Marked-up manuscript**. The red part has been revised according to your comments. The responses to the reviewer's comments are marked in red and presented following:

Answers to Reviewer #2:

1) The new title may be more clear as:

Response of Rhizosphere Bacterial Communities to Near Natural Forest Management and Tree Species Within Chinese Fir Plantations

The author's answer: We are appreciated your suggestion. We have changed the title to **"Response of Rhizosphere Bacterial Communities to Near Natural Forest Management and Tree Species Within Chinese Fir Plantations" (Page 1, Lines 1-2)**.

2) The authors indicate that the Simpson index (not inverse Simpson or 1 - Simpson) is used, which means that the lower the value, the higher the alpha diversity. The data from this alpha diversity index are then in accordance with the other diversity indices used (Figure 2A). This should be corrected before publication.

The author's answer: Thank you very much for your suggestion. After careful confirmation, we agree with you that the Simpson index was used in our article instead of inverse Simpson or 1 - Simpson, and the lower value means the higher alpha diversity. The calculation formula is as follows:

$$D_{simpton} = \frac{\sum_{i=1}^{S_{obs}} n_i(n_i - 1)}{N(N - 1)}$$

S_{obs} : Actual number of OTUs measured;

n_i : Number of sequences contained in i th OTU;

N : Number of all sequences.

However, we can see that the trend of the Simpson index is opposite to that of the Shannon index, Chao1 and ACE from Fig. 2a. **SHX_S has the lowest Simpson's index. In the contrary, Chao1, ACE, and Shannon indices of SHX_S reach the largest.** This is consistent with the result that the Simpson index has an opposite trend to the other indices. Data from the Simpson index were inconsistent with other diversity indices used. For this reason, we chose not to correct this.

3) In the Legend for Figure 9C, correct the spelling for "abbreviations".

The author's answer: Thank you for the suggestion. We have replaced the **"Abbrevaitions"** with **"Abbreviations"** (Page 35, Line 651).

Answers to Reviewer #3:

The paper has been properly revised to my satisfaction

The author's answer: Thank you very much for your suggestions for improving the quality of our articles, and it is a great honor to receive your recognition of this work.

We appreciate for Reviewers' warm work earnestly and hope that the correction will meet with approval. Once again, thank you very much for your comments and suggestions. Look forward to hearing from you!

Yours sincerely,

Aiguo Duan

December 2, 2022

Research Institute of Forestry, Chinese Academy of Forestry, Beijing, China

January 3, 2023

Prof. Aiguo Duan
Chinese Academy of Forestry Research Institute of Forestry
Beijing
China

Re: Spectrum02328-22R2 (Response of Rhizosphere Bacterial Communities to Near Natural Forest Management and Tree Species Within Chinese Fir Plantations)

Dear Prof. Aiguo Duan:

Thank you for submitting your paper to Spectrum. Your manuscript has been accepted, and I am forwarding it to the ASM Journals Department for publication. You will be notified when your proofs are ready to be viewed.

Sincerely,

Kevin R. Theis
Editor, Microbiology Spectrum